# Gliding motility of the diatom *Craspedostauros australis* coincides with the intracellular movement of raphid-specific myosins
Metin G. Davutoglu[1], Veikko F. Geyer [1], Lukas Niese[1], Johannes R. Soltwedel [2], Marcelo L. Zoccoler [2], Valeria Sabatino[1], Robert Haase[2,5], Nils Kröger [1,2,3], Stefan Diez [1,2,4] ✉ & Nicole Poulsen [1] ✉

Raphid diatoms are one of the few eukaryotes capable of gliding motility, which is remarkably fast and allows for quasi-instantaneous directional reversals. Besides other mechanistic models, it has been suggested that an actomyosin system provides the force for diatom gliding. However, in vivo data on the dynamics of actin and myosin in diatoms are lacking. In this study, we demonstrate that the raphe-associated actin bundles required for diatom movement do not exhibit a directional turnover of subunits and thus their dynamics do not contribute directly to force generation. By phylogenomic analysis, we identified four raphid diatom-specific myosins in *Craspedostauros australis* (CaMyo51A-D) and investigated their in vivo localization and dynamics through GFP-tagging. Only CaMyo51B-D but not CaMyo51A exhibited coordinated movement during gliding, consistent with a role in force generation. The characterization of raphid diatom-specific myosins lays the foundation for unraveling the molecular mechanisms that underlie the gliding motility of diatoms.

Cell motility plays a crucial role in the life of unicellular eukaryotes inhabiting aquatic environments. It enables them to actively search for nutrients, evade predators, and locate suitable environments for reproduction and survival. While cilia and flagella-based swimming are well-known forms of motility, some eukaryotes, in particular protists, exhibit other remarkable modes of surface-attached movement, including crawling, rolling, and gliding[1–4]. Gliding motility refers to cell movement on a surface without the involvement of external appendages such as cilia, flagella, or pili[2,5,6]. While gliding motility is widespread in prokaryotes[5], it is limited to certain eukaryotic groups such as algae (e.g., *Chlamydomonas*, diatoms) and Apicomplexans[7] (e.g., *Plasmodium*, *Toxoplasma* and Gregarines). In these cases, the cytoskeleton plays a crucial role in generating a mechanical force, which is transmitted across the plasma membrane to adhesion complexes on the substratum, propelling the cell in the opposite direction to the force exerted intracellularly[2,6,8]. However, between the different groups of organisms, there are significant mechanistic differences in their gliding machineries.

Diatoms are a large group of unicellular algae with silica-based cell walls, found ubiquitously in sunlit aquatic habitats. Amongst the diatoms, only some pennate species that possess a specialized longitudinal slit in their cell wall termed the raphe (and hence being called raphid diatoms), are capable of gliding motility. The raphe is an evolutionary adaption that allows diatoms to colonize and move on submerged surfaces, such as rocks, sand, animals, and other algae[9]. Motile diatoms exhibit impressive velocities, reaching up to 35 $\mu m \cdot s^{-1}$, while navigating intricate, curved trajectories and being capable of quasi-instantaneously reversing their direction[10–13]. While models of the mechanism of diatom gliding date back to 1753, and numerous theories proposed over the years[14–17], the molecular mechanism driving this unusual motility remains unknown.

The prevailing model relies on the concept of an actomyosin-based adhesion motility complex (AMC)[17–19] and is based on the presence of two bundles of actin filaments positioned beneath the plasma membrane adjacent to the raphe opening (termed raphe-associated actin; RA-actin)[20]. According to the AMC model, there exists a physical connection between

[1]B CUBE - Center for Molecular Bioengineering, TUD Dresden University of Technology, Dresden, Germany. [2]Cluster of Excellence Physics of Life, TUD Dresden University of Technology, Dresden, Germany. [3]Faculty of Chemistry and Food Chemistry, TUD Dresden University of Technology, Dresden, Germany. [4]Max Planck Institute of Molecular Cell Biology and Genetics, Dresden, Germany. [5]Present address: Center for Scalable Data Analytics and Artificial Intelligence, Faculty of Mathematics and Computer Science, Leipzig University, Leipzig, Germany. ✉e-mail: stefan.diez@tu-dresden.de; nicole.poulsen@tu-dresden.de

the RA-actin through a continuum of biomolecules spanning the plasma membrane to the extracellular polymeric substances (EPS) secreted from the raphe, which enable adhesion and traction on the surface. Myosin motor proteins moving along the RA-actin are hypothesized to exert mechanical force in the AMC which is then transmitted to the substratum, enabling cell movement in the opposite direction of myosin movement[17–19]. So far, the most compelling evidence to support this model comes from experiments using free-floating microscopic beads. Beads that adhere to the raphe can display bi-directional movement on both stationary and gliding cells, with velocities similar to the velocity of the diatom gliding[17,21–23]. Recently, Gutiérrez-Medina and colleagues observed two distinct patterns of bead movement along the raphe: (i) smooth motion lasting several seconds at low velocities and (ii) intermittent, jerky motion with short (<100 ms) episodes of heightened velocity and acceleration[13]. These two patterns were found to be present during both uni-directional and bi-directional bead movement, and are attributed to a model in which cells alternate between smooth, sustained movement driven by molecular motors and abrupt, fast movement due to the elastic snapping of EPS strands. Although experiments utilizing actin and myosin inhibitors have demonstrated reversible inhibition of gliding motility[19], direct evidence establishing actin and myosin as the principal force generators remains to be found.

Within the realm of eukaryotes, myosin motor proteins exhibit significant divergence, encompassing over 70 distinct classes, based on phylogenetic analysis of their motor domain sequences and domain architecture[24,25]. Recent investigations involving diatom genome and transcriptome sequencing projects have shed light on the phylogenetic relationships of myosins within this taxonomic group. Each diatom species possesses a repertoire of ten to eleven different myosins that fall into one of five distinct classes, and one or more myosins that remain unclassified and are referred to as 'orphan myosins'[25]. To date, no diatom myosin has been functionally characterized. Recent advances in genetic engineering of the raphid diatom *Craspedostauros australis*[26–28] have enabled new approaches to investigate the proposed role of myosins as force generators during diatom gliding. In this study, we aim to test this hypothesis by identifying raphid-specific myosins through phylogenomic analysis and by relating their intracellular dynamic behavior to cell movement.

## Results

### GFP-labeling allows live-cell imaging of the actin cytoskeleton in *C. australis*

Previously the actin cytoskeleton of raphid diatoms has been visualized exclusively in fixed cells stained with fluorescently labeled phalloidins[19,29–31]. To study the structure and dynamics of actin in living cells, we generated *C. australis* cell lines expressing N-terminally GFP-tagged actin. We identified four copies of the actin gene in the *C. australis* genome assembly that differ in the size and position of the single intron, but share almost identical amino acid sequences (Supplementary Figs. 1–3). Screening of the resulting cell lines identified clones with a high ratio of GFP-labeled actin that were used for the precise localization of the actin cytoskeleton, and clones with lower amounts of GFP-labeled actin ('speckled') that were used for quantitative analysis of actin dynamics. The latter was crucial to (i) ensure that fluorescent labeling of the actin bundles is discontinuous, thereby enabling the detection of actin dynamics, and (ii) to minimize potential artifacts known to be caused by GFP-tagging of actin that could compromise its function[32,33]. Motile cells were observed in all clones irrespective of the abundance of GFP-actin, indicating that the GFP tag did not inhibit cell motility. Motility assays of populations of wild-type and GFP-actin-expressing cells showed no striking differences in the velocity of motile cells (Supplementary Fig. 4).

Confocal fluorescence microscopy confirmed the presence of two RA-actin bundles that are continuous along the longitudinal axes of the two opposite valves, which together with the girdle band, form the diatom frustule (Fig. 1a–e). There is a distinct separation of the two RA-actin bundles in the center of the cell, beneath the central nodule. At the cell apices, the RA-actin bundles follow the shape of the valves (Fig. 1b, d) and do not extend into the girdle band region (Fig. 1c, e). Imaging of the cell apices

revealed that the two RA-actin bundles appear to be continuous, forming narrow loops each with a diameter of ~250 nm (Fig. 1f, g). In addition, a pronounced ring of actin filaments is associated with the girdle bands, and an extensive network of actin was observed in the perinuclear region of the cell, which corresponds to the position of the Golgi complex (Fig. 1e).

### Actin dynamics do not contribute to the generation of force for cell gliding

To investigate the possible involvement of actin dynamics (for example via filament treadmilling) in gliding motility, we performed time-lapse dual-color total-internal-reflection fluorescence microscopy (TIRFM) on both mobile and stationary cells expressing GFP-tagged RA-actin (Fig. 1h–k). This technique enabled the simultaneous imaging of the GFP-tagged RA-actin and chloroplast autofluorescence in the evanescent field of TIRFM close to the coverslip (Fig. 1h). We developed an image analysis pipeline that utilizes tracking of chloroplast autofluorescence as a proxy for cell movement. By effectively subtracting the cell movement from the kymograph (space-time plots of the fluorescence intensities along the direction of the raphe), we were able to relate the intracellular movement of fluorescently labeled actin (or myosin, see below) with respect to cell movement (Supplementary Fig. 5, Methods).

A technical limitation encountered during TIRFM imaging of the GFP-labeled RA-actin bundles arises from their spatial distance from the coverslip. This distance is subject to variations as (i) the RA-actin bundles follow the curved shape of the cell wall (Fig. 1b, d)[27] and (ii) the EPS layer varies in thickness during gliding motility. Consequently, it is challenging to visualize the entire length of the bundles during cell movement. Nonetheless, as the cells glided across a coverslip, several micrometer-long sections of the GFP-actin bundles were visible along the longitudinal axis of the cell (Fig. 1i, Supplementary Fig. 6a, Supplementary Movies 1, 2). Kymograph analysis revealed that there was no discernable net movement of the GFP labeled RA-actin relative to the cell in gliding or stationary cells within time intervals of 25–30 s (example kymographs in Fig. 1j, k, Supplementary Fig. 6b). These observations indicate that there is no directional turnover of actin subunits within the RA-actin bundles, and thus we proceeded with the assumption that actin dynamics do not contribute to force generation for gliding.

### Phylogenomic analysis reveals four *C. australis* myosins in a raphid-specific clade

Myosins are currently classified into 79 classes based on phylogenetic analysis of their motor domain sequence[25]. As gliding motility is restricted to raphid diatoms, we performed a phylogenomic analysis on all publicly available diatom myosin sequences to determine which myosins are specific to raphid diatoms. BLAST analysis of the *C. australis* genome assembly[27,28] revealed twelve putative myosin sequences (Supplementary Figs. 7 and 8), a number which is similar to other sequenced diatoms (*Phaeodactylum tricornutum*: 10; *Thalassiosira pseudonana*: 11)[34]. The phylogenomic analysis was performed using the predicted motor domains of 309 diatom myosins from 52 diatom species (Supplementary Data 1) and corroborated previous findings distinguishing five diatom myosin classes (Class 29, 47, 51, 52 and 53) as well as between one and four orphan, unclassified myosin sequences[24,25,34,35].

Four of the *C. australis* myosins, which we named CaMyo51A-D (all belonging to myosin Class 51), were exclusively found in raphid diatoms (Supplementary Fig. 7, clade highlighted by blue shade). CaMyo51A-D possess canonical myosin domains (motor, neck, tail) and contain essential motor domain motifs including the purine binding loop, Walker A motif, switch 1/2 regions, actin-binding site, Src homology 1 helix, and possess variable numbers of IQ motifs in the neck region (Fig. 2a, Supplementary Fig. 9). The tail domains of these four myosins are relatively short (276-417 amino acids), do not contain any additional protein domains, and are predicted to contain coiled-coil regions suggesting they are dimeric in vivo. Based on the phylogenomic distance of these four myosins to previously described *P. tricornutum* myosins suggests homology of: CaMyo51A – PtMyoG, CaMyo51B/C – PtMyoC/A, CaMyo51D-PtMyoE[34].

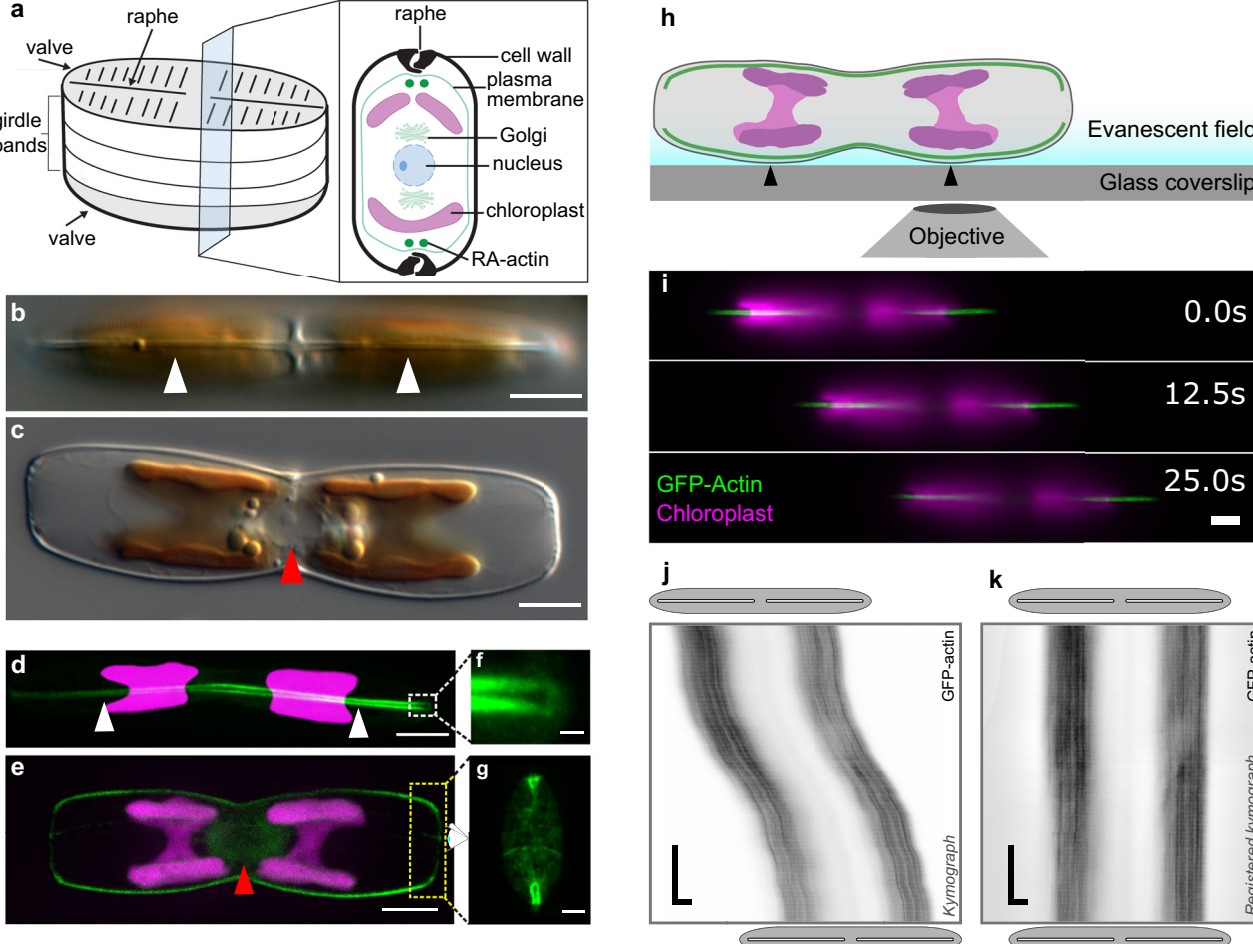

**Fig. 1 | Analysis of actin structure and dynamics. a** Schematic representation of a pennate cell wall, showing a cross section though the cell and intracellular structures. Differential interference contrast (DIC) light microscopy images of *C. australis* wild type cells in valve view (**b**, white arrowheads point to the raphe slit) and girdle view (**c**, red arrowhead points to the perinuclear region). The chloroplast has a golden-brown pigmentation. Confocal fluorescence microscopy images of live cells embedded in agarose expressing GFP-actin. Z-projection of optical sections of the proximal raphe slit adjacent to the glass coverslip **d** (white arrowheads point to the same position as in b) and the midline section of the cell **e** (red arrowhead pointing to perinuclear actin). Magnified loops of raphe-associated GFP-actin at the distal ends of cells embedded in agarose in raphe view (**f**, magnification of white dotted box in **d**) and in oblique view (**g**, approximate location of yellow dotted box shown in **e**, rotated

90 degrees away from the viewer, shown for a different upright cell). **h** Schematic of live cell TIRFM experimental setup. Cells adhere and glide with one set of actin bundles (green) and two chloroplasts (magenta) in the evanescent field close to the coverslip surface. Black arrowheads indicate points where the cell wall is in close proximity to the coverslip. **i** Montage generated from dual-color TIRFM images showing the position of a gliding cell at three time points. Kymograph analysis of GFP-actin movement (in black) using raw **j** and registered **k** GFP-actin data from **i**. Gray ellipses and white slits above and below kymographs approximate the positions of the cell body and raphe openings at the beginning and end of imaging, respectively. (green – actin; magenta – chloroplast autofluorescence; scale bars: **b**–**e**, **i**: 5 μm, **f**: 0.5 μm, g: 2 μm, **j**, **k**: horizontal 5 μm, vertical 5 s).

## CaMyo51A-D are distributed along the longitudinal axis of the valve and in the perinuclear region

To explore the involvement of CaMyo51A-D in generating force for gliding motility, we expressed them as C-terminally tagged GFP-fusion proteins. First, we visualized their intracellular localization by confocal fluorescence microscopy in cells embedded in agarose, where gliding motility was suppressed (Fig. 2b–i). All four GFP-tagged myosins (i) were localized along the longitudinal axis of each valve with varying degrees of continuity in the raphe region, (ii) exhibited a high abundance around the perinuclear region, and (iii) were distributed throughout the cytoplasm in a mesh-like network similar to the distribution of GFP-tagged actin.

CaMyo51A-GFP was typically seen localized along the longitudinal axis of both valves in two interrupted linear patterns, which extended to the cell apices and like the RA-actin bundles diverged near the center of the valve where the central nodule is located. In contrast, CaMyo51B-GFP was typically only observed in discrete fluorescent regions along the longitudinal axis of the valve, and it was not possible to distinguish two separate linear

patterns. Based on the position of the chloroplasts, the discrete regions of CaMyo51B-GFP fluorescence were close to the curved regions of the valve, and likely closer to the cover glass (see Fig. 1h). CaMyo51C-GFP was typically seen in two dense linear patterns along the longitudinal axis of both valves. Like CaMyo51A-GFP, the distance between these linear patterns was seen to diverge near the central nodule. CaMyo51D-GFP was also seen in two linear patterns along the longitudinal axis both valves, but additionally showed distinct accumulations at the cell apices.

## TIRFM reveals dynamic inhomogeneities of CaMyo51A-D along the longitudinal axis of the cell

To investigate the dynamic behavior of the myosins, we performed live cell dual-color time-lapse TIRFM imaging on both mobile and stationary cells with GFP fusion proteins of CaMyo51A-D (Figs. 3–6, Supplementary Fig. 10). For all four myosins we observed a rather uniform distribution of GFP-fluorescence along the entire longitudinal axis of each valve, closely matching the position of the RA-actin bundles. In addition, we observed

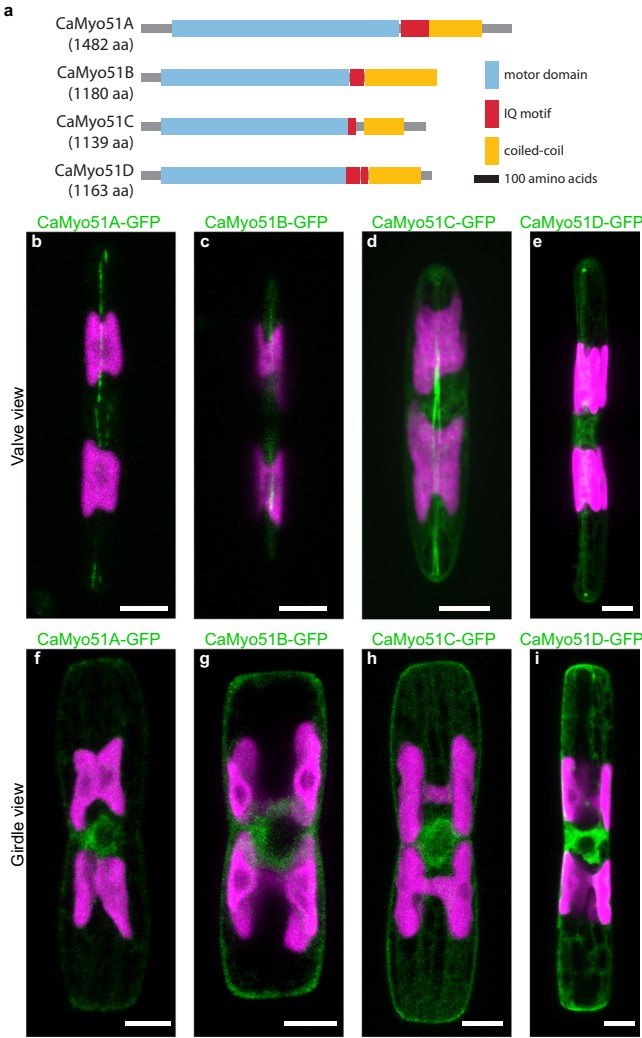

**Fig. 2 | Schematic primary structures and intracellular localization of CaMyo51A-D. a** Schematic domain structure of the four myosins investigated in this study. Scale bar: 100 amino acids. **b–i** Localization of myosin-GFP fusion proteins. Each image is a maximum z-projection of the confocal fluorescence microscopy stack of the indicated GFP-tagged myosin. GFP-tagged myosin fluorescence shown in green and chloroplast autofluorescence in magenta. Scale bars: 5 μm.

some inhomogeneities in the GFP-fluorescence signal, manifesting as individual or multiple distinct spots. Specifically, these spots were observed for (i) CaMyo51A-GFP in predominantly stationary (or slow-moving) cells, (ii) CaMyo51B-GFP and CaMyo51C-GFP exclusively in gliding cells, and (iii) CaMyo51D-GFP both in stationary and gliding cells. Despite these differences in the localization and activity of these four GFP labelled myosins, their expression did not lead to any striking changes in the velocity of motile cells compared to wild type cells (Supplementary Fig. 4).

According to the AMC hypothesis, we would expect a myosin contributing to cell motility (i) to be active on the RA-actin bundles, (ii) exhibit little to no movement relative to the substratum, and (iii) move towards the trailing end of the moving cell. To investigate which myosins show such behavior, we analyzed the dynamic behaviour of individual spots of myosin-GFP fusion proteins in continuously moving cells. CaMyo51A-GFP spots occasionally showed slow, non-coordinated movement (Fig. 3a, Supplementary Movie 3). Spots of CaMyo51B-GFP, CaMyo51C-GFP, and CaMyo51D-GFP were frequently observed to move persistently and in a coordinated manner towards the trailing end of the cell (Fig. 3b-d, Supplementary Movies 4-6). The trajectories of spots of CaMyo51B-GFP, CaMyo51C-GFP, and CaMyo51D-GFP were parallel to the long axis of the

valves, crossed the midpoint of the valve, and often exhibited nearly constant velocities over extended time periods.

## Intracellular motility of CaMyo51B-D, but not CaMyo51A, coincides with cell motility

We further quantified the relationship between myosin and cell motility using our image analysis pipeline described above. We found that the majority of CaMyo51A-GFP spots moved in long, slow runs in both directions within stationary and slowly-moving cells (Supplementary Fig. 6, Supplementary Movies 7-9). CaMyo51A-GFP spots often passed one another, moving in the opposite direction and their mean velocity was typically slower than the gliding velocity of the cells (Fig. 4). Moreover, the spots did not strictly follow the centerline of the valve, suggesting that CaMyo51A-GFP is not exclusively moving on the RA-actin bundles. In contrast, the CaMyo51B-GFP, CaMyo51C-GFP, and CaMyo51D-GFP spots moved exclusively in opposite direction to cell movement. The absolute values of the associated myosin velocities (ranging up to 12 μm s$^{-1}$) always exceeded the cell velocities (ranging up to 4 μm s$^{-1}$) (Fig. 4).

If intracellular myosin motility were to be directly involved in propelling cell motility, one would further expect that changes in myosin velocity coincide with changes in cell velocities upon changes in the motility state of moving cells. We indeed observed this relationship for (i) starting cells (i.e. cells transiting from a resting state to a moving state, Fig. 5a), (ii) stopping cells (i.e. cells transiting from a moving state to a resting state, Fig. 5b), and (iii) cells reversing their direction (Fig. 5c). In all cases, the motility of CaMyo51B-GFP (which exhibited the most consistent behavior among the myosins) showed fast coordinated movement when gliding started, abrupt halting when gliding stopped and quasi-instantaneous directional reversal when gliding switched direction (Fig. 5, Supplementary Movies 10–12). Similar behaviors were observed for CaMyo51C-GFP and CaMyo51D-GFP spots, though they occasionally moved in opposite directions (either towards the center or towards the apices of the cell valve) in the leading and trailing halves of the same cell (Fig. 6, Supplementary Movies 13–15). The latter cases were rarely observed during smooth, sustained gliding but predominantly in slow-moving cells exhibiting frequent fluctuations in their velocity and/or their direction of cell movement.

## Discussion

In this study, we investigated the involvement of actin dynamics and four raphid-specific myosins (CaMyo51A-D) in the gliding motility of the diatom *C. australis*. Our results demonstrate that diatom gliding does not rely on a direct force contribution from actin treadmilling, and is thus markedly different from the mechanisms of lamellipodia-driven crawling motility or apicomplexan gliding motility that both require actin polymerization[36,37]. Because of the rigid diatom cell wall, movement driven by actin-based cell membrane protrusions would need to extrude through the raphe as proposed in the actin pin hypothesis[16,38]. However, our actin localization experiments showed no evidence of actin-based cell membrane protrusions through the raphe and beyond the cell wall. Additionally, many diatom species have been shown to lack profilins and the Arp2/3 complex[39], both of which are essential regulators of actin branching and nucleation required for membrane protrusions in other eukaryotes.

The precise architecture of the RA-actin bundles, such as the length of individual actin filaments and their polarity within a single bundle remains unknown. Consequently, how these factors may contribute to the bi-directional diatom gliding mechanism remains an open question. Based on our observation that both cells and intracellular myosins can quasi-instantaneously reverse their direction, several possibilities emerge: (i) the two actin bundles are antiparallel and unidirectional, (ii) both actin bundles are bidirectional, or (iii) at least one of the myosins is capable of moving towards the minus-end of the actin filaments. Notably, our investigation into the RA-actin bundles revealed loop-like structures at the cell apices connecting the two actin bundles, suggesting the intriguing possibility that they may form a single continuous bundle. In this case, such a continuous

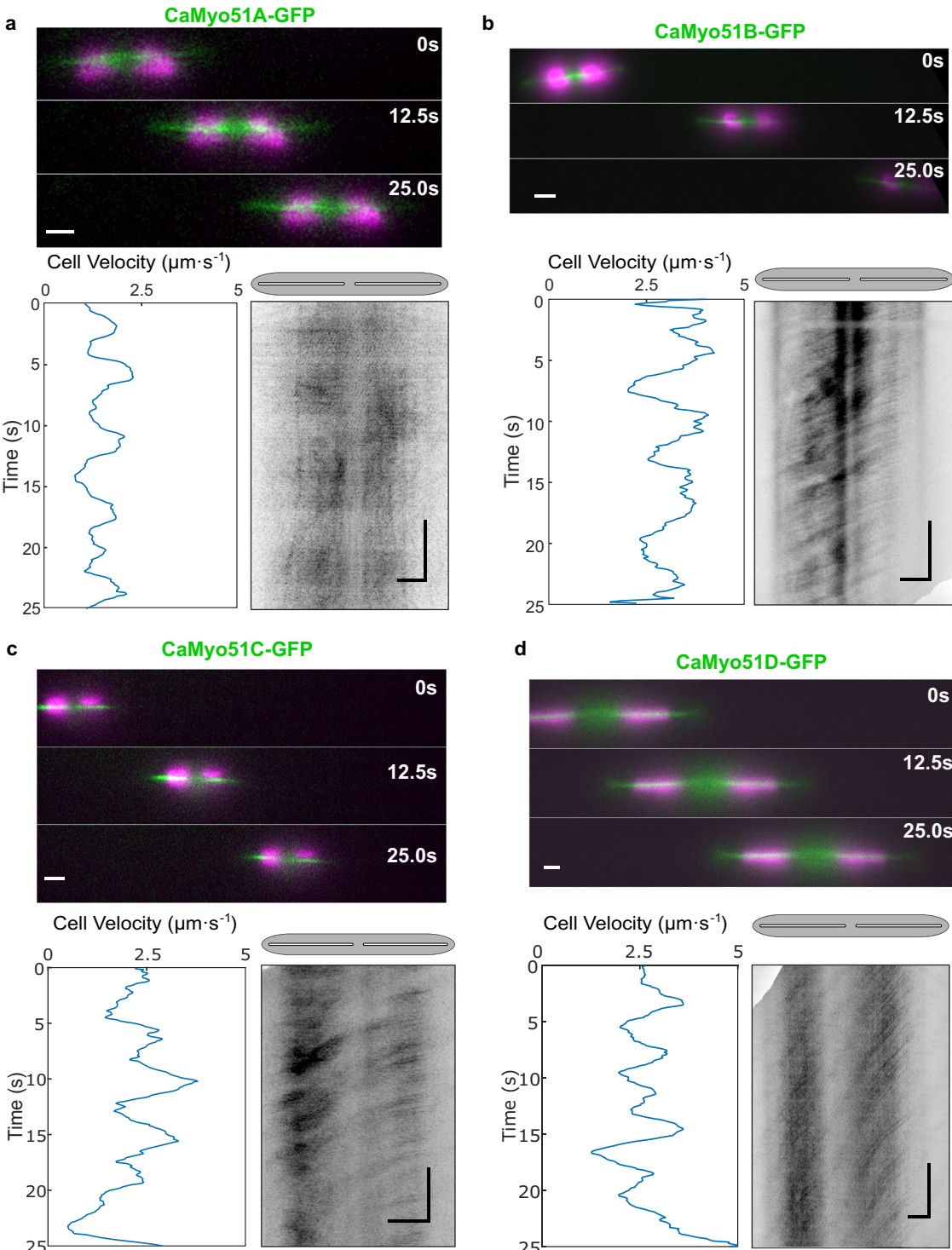

**Fig. 3 | Dynamic localization of CaMyo51A-D in cells during, smooth sustained gliding.** Analysis of 25 s time-lapse segments of gliding cells expressing **a** CaMyo51A-GFP, **b** CaMyo51B-GFP, **c** CaMyo51C-GFP, and **d** CaMyo51D-GFP. (Upper panels) Montages showing the position of cells at 12.5 s intervals (GFP in green, chloroplast autofluorescence in magenta, scale bars: 5 μm). (Lower panels, left) Cell velocity as a function of time, generated from chloroplast tracking data. (Lower panels, right) Registered kymographs generated from GFP-channel data (black) show the movement of myosins relative to the cell. Gray ellipses and white slits above kymographs approximate the positions of the cell body and raphe openings, respectively. (Scale bars: horizontal = 5 μm, vertical = 5 s).

track would eliminate the need for secondary transport of myosins once the end of a bundle is reached, even if the individual actin bundles are not bidirectional.

Our results demonstrate that CaMyo51B-D, but not CaMyo51A, engage in fast, coordinated movement in the opposite direction to cell movement. This inverse relationship between the direction of myosin movement and cell gliding aligns with the hypothesized force coupling mechanism within the AMC model[17]. Although CaMyo51B-D displayed comparable kinetic behaviors, which may indicate some level of redundancy, we cannot rule out additional isoform-specific functions, which may play a role during the generation of complex trajectories, directional changes, and intracellular transport.

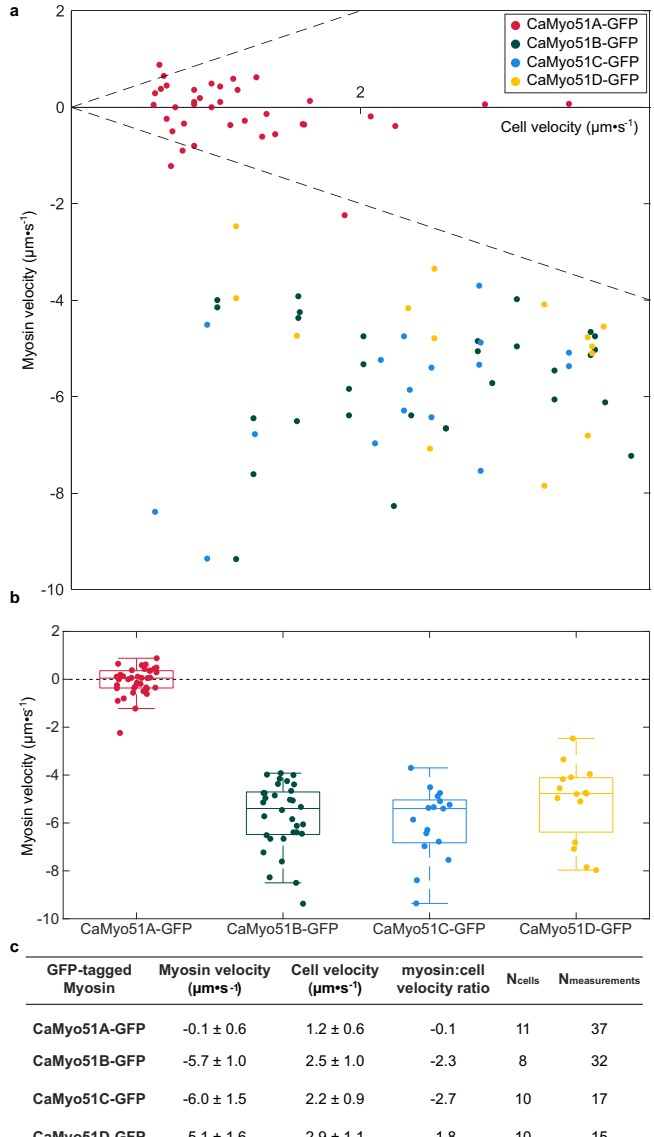

**Fig. 4 | Quantitative relationship between cell velocities during smooth, sustained gliding and intracellular GFP-tagged myosin velocities. a** Mean myosin velocity relative to the moving cells (each data point signifies one measurement in one half of the cell during a given time window, see Materials and Methods) as function of mean cell velocity (during the same time window). Dashed line indicates the equality of myosin and cell velocity. **b** Box plots showing distribution of myosin velocities for four GFP-tagged myosins analyzed in this study (central line—median value, box edges—25th and 75th percentiles, whiskers—minimum and maximum values, circles—sample outliers). **c** Myosin velocities (mean +/− SD) and corresponding mean cell velocities (mean +/− SD) along with their ratio, as well as, number of cells ($N_{cells}$) investigated and total measurements ($N_{measurements}$, with each measurement representing an average of tens to hundreds of traces of myosin spots in the respective kymograph) performed.

| GFP-tagged Myosin | Myosin velocity (µm•s⁻¹) | Cell velocity (µm•s⁻¹) | myosin:cell velocity ratio | $N_{cells}$ | $N_{measurements}$ |
|---|---|---|---|---|---|
| CaMyo51A-GFP | -0.1 ± 0.6 | 1.2 ± 0.6 | -0.1 | 11 | 37 |
| CaMyo51B-GFP | -5.7 ± 1.0 | 2.5 ± 1.0 | -2.3 | 8 | 32 |
| CaMyo51C-GFP | -6.0 ± 1.5 | 2.2 ± 0.9 | -2.7 | 10 | 17 |
| CaMyo51D-GFP | -5.1 ± 1.6 | 2.9 ± 1.1 | -1.8 | 10 | 15 |

In their natural habitat, diatoms must navigate their surroundings, respond to chemical gradients, and migrate towards or away from specific cues (e.g. light, nutrients, toxins)[40–43]. Remarkably, they accomplish these tasks with a high degree of temporal and spatial control[44]. Individual diatom species can vary their cell velocities by one order of magnitude, are capable of complex movements in circular, spiral, straight, and even sometimes sigmoidal trajectories, and are able to reverse their cell direction quasi-instantaneously[11,12,45]. While our study has yet to investigate the behavior of GFP-tagged CaMyo51B, -C, and –D, simultaneously within a single cell line or over longer cell trajectories, it is conceivable that the

distinct behaviors observed for each myosin contribute to disrupting symmetry and distributing forces along the entire raphe. This, in turn, may pave the way for changes in the direction and velocity of cell movement. Consequently, by modulating the activity of a combination of myosin isoforms, diatoms may be capable of generating forces in both directions along the raphe, ultimately underpinning the intricate trajectories witnessed in the gliding diatoms.

During smooth, sustained gliding, multiple spots of each CaMyo51B, CaMyo51C and CaMyo51D move at uniform velocities, suggesting a high degree of myosin cooperativity along the entire length of the cell. Theoretically, the force generated by a single myosin would be sufficient to overcome the viscous drag of a diatom moving in aqueous solution; according to Stoke's law the viscous drag of a cylinder with half-sphere ends and a diameter of 6 µm moving through water at 4 µm s⁻¹ is less than 1 pN. Nevertheless, recent research has shown that diatoms are capable of generating forces far greater (~800 pN) than those required to propel cell movement, which is consistent with a large number of myosin motors being simultaneously involved in force generation[45]. This becomes particularly relevant when considering the natural habitat of raphid diatoms, where they must force their way through fine-grained sediments laden with obstacles, which is in stark contrast to the controlled environment of a smooth laboratory glass slide. Furthermore, when embedded within a dense biofilm matrix, diatoms must exert sufficient mechanical forces to overcome the elastic resistance of the biofilm EPS. It is therefore likely that cooperative myosin activity enables (i) the generation of force large enough to overcome obstacles and (ii) the coordination of directional forces exerted at multiple raphe openings to establish, maintain, or reverse the direction of gliding. We hypothesize that the cooperative activity of myosin may be orchestrated through chemical signaling pathways (such as phosphorylation or calcium ions)[46] or physical interactions (such as oligomerization of their tail domains)[47,48].

We consistently observed that the velocity of cell movement was slower (up to 50% and less) compared to the velocity of the GFP-tagged myosins. This finding strongly suggests mechanical compliance in the motility machinery. According to the AMC model, diatom gliding requires the transmission of a force generated by the intracellular cytoskeleton across the plasma membrane and through the raphe slit to the extracellular substratum via adhesive EPS strands. Consequently, the loss in velocity, which indicates a loss in force transduction from the actomyosin complex to the substratum, could be caused by (i) inefficient force transfer through the plasma membrane and raphe slit (similar to IFT trains that drive gliding motility in Chlamydomonas[8] and to slippery motor anchorage in lipid bilayers[49]), or (ii) the mechanical properties of the EPS strands, which exhibit high tensile strength and elasticity attributed to the presence of large modular proteins[50]. Some of these EPS strands, in particular those not connected to the AMC, may exert elastic or frictional counterforces potentially causing a dissipation of energy during force transmission[50]. Recent findings have begun to unveil the molecular composition of the diatom EPS, identifying two exceptionally large modular proteins in the *C. australis* adhesive trails (CaTrailin4; ~900 kDa and Ca5609; >1 MDa)[28], which may contribute to these counterforces, and a mucin-like protein (CaFAP1)[27] that may contribute to reducing the frictional resistance between the cell and the substratum. Furthermore, these very large proteins, approximately 2 µm in length, are sufficiently long to span the physical distance created by the thickness of the raphe slit (300–500 nm in *C. australis*), allowing them to bridge the gap between the intracellular cytoskeleton and underlying surface. Consequently, they could serve a dual function as both a force transducer and adhesive. However, how these components, and others, contribute to mediating interactions between the intracellular cytoskeleton, the cell surface, and the substratum beneath remain to be investigated.

Occasionally, spots of CaMyo51C and CaMyo51D moved in a fast and coordinated manner toward the center of a gliding cell in both the leading and trailing halves. This phenomenon typically coincided with an abrupt change in the direction of cell movement or comparatively low gliding

**Fig. 5 | Dynamic localization of CaMyo51B in cells during changes in their motility state.** Analysis of 20–30 s time-lapse segments of CaMyo51B-GFP expressing cells during **a** starting, **b** stopping, and **c** reversing. (Left panels) Montages showing the position of cells at 5 s intervals (GFP in green, chloroplast autofluorescence in magenta, scale bars: 5 μm). (Middle panels) Cell velocity as a function of time, generated from chloroplast tracking data. (Right panels) Registered kymographs generated from GFP-channel data (black) showing movement of myosins relative to the cell. The small offsets between starting/stopping/reversing of cell and myosin activity (dotted horizontal lines between middle and right panels) likely indicate mechanical compliance in the motility machinery. Gray ellipses and white slits above kymographs approximate the positions of the cell body and raphe openings, respectively. (Scale bars: horizontal = 5 μm, vertical = 5 s).

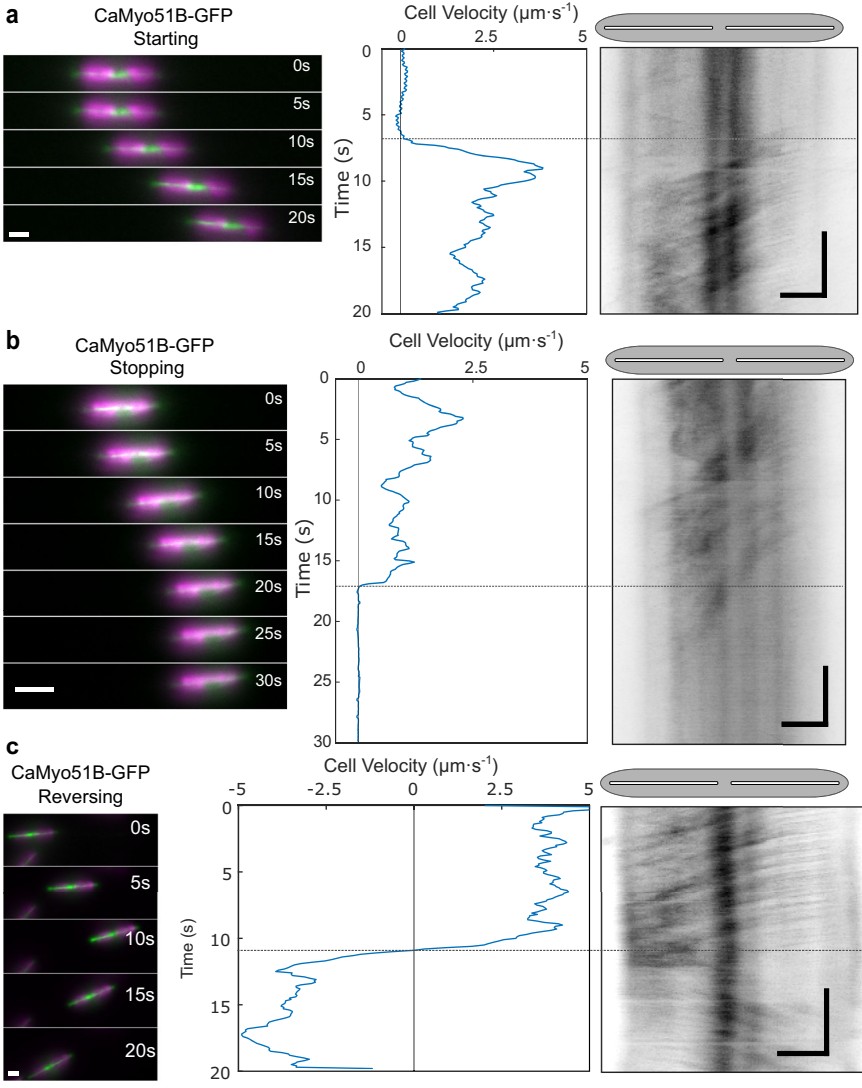

velocities (<1.5 μm·s⁻¹), suggesting that these myosins can generate opposing forces in each half of the cell. Opposing forces play an important role in many cytoskeletal processes including bidirectional transport of vesicles, cell size determination, and actin ring formation in cytokinesis, which rely on similar tug-of-war systems between motors[51–53]. In vitro experiments have also shown that groups of molecular motors pulling in opposite directions can cause spontaneous symmetry breaking[54] and distinct modes of slow and fast movements, as well as sharp transitions between these modes[55]. The observation of counteracting force production by both CaMyo51C and CaMyo51D corroborates a recent report, which has indirectly demonstrated the existence of such opposing forces in diatom motility[18,56]. By distributing small particles on the transparent diatom biofilm matrix, Harbich elegantly demonstrated that these forces can be observed as a compression in the surrounding biofilm matrix when a diatom stops moving, causing a strain to be drawn towards the proximal raphe ends[56]. The opposing activity at the two raphe openings creates tension that stalls cell movement. Once the two raphes cease working against each other, uniform gliding motility resumes.

CaMyo51A displayed a single mode of very slow, non-cooperative myosin movement without any observed coordination with overall cell movement. This movement pattern is reminiscent of the behavior of myosins engaged in intracellular transport, such as class V myosins. The high duty ratio of class V myosins enables them to move processively during force generation, resulting in long, slow runs during which the myosin does

not detach from actin[57,58]. Such intracellular transport may contribute to gliding by delivering vesicles containing EPS to the raphe. Golgi-derived vesicles containing 'fibrillar polysaccharides' have been observed near the RA-actin bundles in numerous motile diatoms[17]. These vesicles are believed to fuse with the plasma membrane and release their contents into the raphe, although the exact fusion sites and their role in gliding motility are still debated[17,59]. Nevertheless, current evidence suggests that vesicle delivery alone does not drive motility. The observed particle streaming along the raphe indicates a more complex intracellular control of movement[17], likely involving the coordinated movement of CaMyo51B-D observed in this study.

Our study provides a crucial link to understanding the hypothesized diatom 'adhesion motility complex'. However, we cannot exclude the potential contributions of other cellular mechanisms. For example, the recently described 'raphan synthase' model[59] suggests that a cellulose-synthase-like complex (termed raphan synthase) embedded in the membrane produces polysaccharide fibrils (termed raphans) that enter the raphe and swell to generate the force for movement. In case such a mechanism plays a role in force generation, the coordinated myosin movement observed in our study may be involved in moving the proposed raphan synthase within the membrane. Furthermore, the exact location and mechanism of adhesive strand secretion needs to be determined and it will be important to to explore whether the proposed continuum between these strands and the intracellular RA-actin bundles exists.

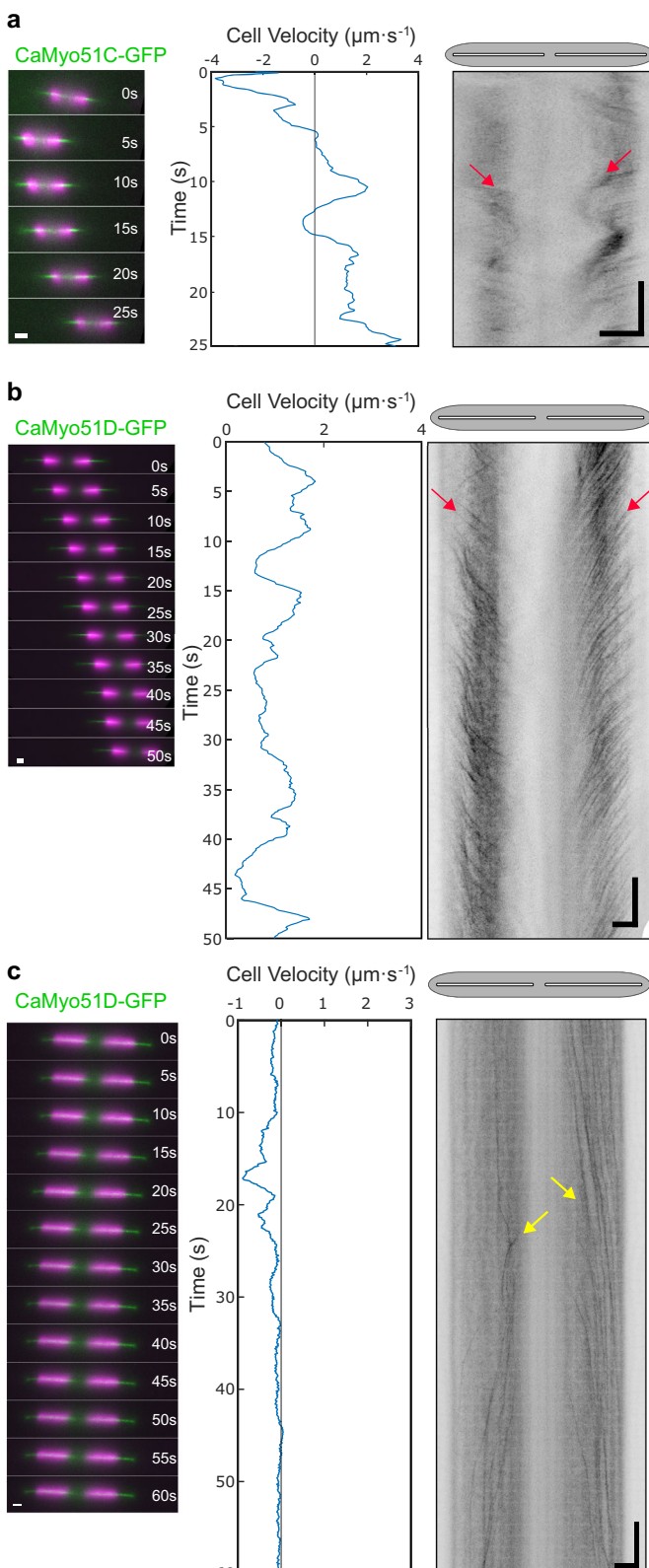

**Fig. 6 | Multi-directional movements of CaMyo51C-GFP and CaMyo51D-GFP.** Analysis of 25-60 s time-lapse segments of CaMyo51C-GFP and CaMyo51D-GFP expressing cells showing **a** CaMyo51C-GFP movement from apices to the cell center (red arrows) in a cell exhibiting repeated changes in gliding velocity, **b** CaMyo51D-GFP movement from apices to the cell center (red arrows) in a cell exhibiting slow unidirectional gliding, and **c** CaMyo51D-GFP movement from the center to the cell apices (yellow arrows) in a mostly stationary cell. (Left panels) Montages showing the position of cells at 5 s intervals (GFP in green, chloroplast autofluorescence in magenta, scale bars: 5 µm). (Middle panels) Cell velocity as function of time, generated from chloroplast tracking data. (Right panels) Registered kymographs generated from GFP-channel data (black) showing movement of myosins relative to the cell. Gray ellipses and white slits above kymographs approximate the positions of the cell body and raphe openings, respectively. (Scale bars: horizontal = 5 µm, vertical = 5 s).

or if they are capable of fulfilling this role interchangeably. Beyond these results, our work opens avenues for further exploration of myosin functions in other eukaryotes as well as for the design of smart synthetic mechanosystems inspired by biological organisms. As such, it would be intriguing to design molecular transport, manipulation, and sorting devices that combine the distinct diatom properties of large forces, high velocities, and rapid directionality switching.

## Methods

### Diatom strains and culture conditions

*Craspedostauros australis* (Cox, CCMP3328) was grown in artificial seawater medium, EASW[60] or ASW[61] at 18 °C under at a light intensity between 40 and 60 µmol photons·m$^{-2}$·s$^{-1}$, using a cool white light source and unless otherwise stated, maintained with a 14/10-hour light/dark cycle. All cell lines were subcultured regularly (every 7–10 days) to maintain cells in a logarithmic stage of growth. All mutant cell lines were maintained in a medium supplemented with 450 µg/mL nourseothricin (clonNAT) antibiotic (Jena Biosciences, Jena, Germany).

### Identification of myosin and actin genes in *C. australis*

To identify putative *C. australis* myosin genes a tBLASTn protein homology search using available myosin and actin proteins from *Phaeodactylum tricornutum*[34] was performed against the *C. australis* genome assembly[27,28]. All protein sequences were validated for the presence of conserved myosin and actin domains using an Interpro search (https://www.ebi.ac.uk/interpro/).

### Multiple sequence alignment and phylogenomic analysis of diatom myosin sequences

To identify diatom myosin sequences, the head domain of *C. australis* myosin A (CaMyo51A) was used to conduct a BLAST search (E-value ≤ 10$^{-15}$) against available diatom genome and transcriptome databases[62–64] (Supplementary Data 1). The retrieved predicted protein sequences were manually curated to remove duplicates and include only those that contain motifs and domains that are indicative of a myosin head domain (i.e. P-loop, switch-I region, acting binding site, and IQ motifs). A multiple sequence alignment of the trimmed head domains (from the P-Loop to the to the first IQ motif) was performed in Clustal Omega with the default parameters (Supplementary Data 2). The output PHYLIP alignment file was used for a maximum likelihood phylogenetic analysis using IQ-TREE v2.3.4[65]. To assess branch supports, the analysis employed 1000 ultrafast bootstrap replicates[66,67] and 1000 replicates of the SH-aLRT algorithm[68]. The resulting tree file was visualized using the R package ggtree[69] and nodes satisfying the conditions of UFBoot ≥95 and SH-aLRT ≥80 were highlighted.

### Confirmation of myosin gene structure

The full-length gene models of *C. australis* CaMyo51A, CaMyo51B, CaMyo51C, and CaMyo51D were confirmed using RACE and RT-PCR. *C. australis* cDNA and genomic DNA were prepared as previously described[27].

Taken together, our findings suggest that the action of raphid-specific myosins plays a role in diatom gliding motility. Nevertheless, the precise contribution of each myosin to driving cell movement has yet to be determined. It will also be of interest to track the myosins described in this work simultaneously in the same cell. The results of such experiments could reveal whether these myosins work in concert to drive smooth, sustained gliding,

Briefly, for gDNA isolation, the cells were frozen in liquid $N_2$ and ground to a fine powder using a mortar and pestle and then resuspended in lysis buffer (7 M urea, 1% SDS, 10 mM EDTA, 0.1 M Tris–HCl pH 7.5, 0.1 M β-mer-captoethanol) and incubated at 65 °C for 30 min. The resulting cell lystate was extracted four times with phenol:chloroform:isoamly alcohol (25:24:1) and then once with chloroform:isoamyl alcohol (24:1). DNA was pre-cipitated using 0.1 volume of 3 M acetate and 2.5 volumes of ethanol. For cDNA synthesis, mRNA was isolated using the Dynabeads mRNA DIRECT Kit (ThermoFisher Scientific) and reversed transcribed into cDNA using Superscript IV Reverse Transcriptase (ThermoFisher Scientific). To amplify the cDNA ends, two nested RACE PCRs (primer combinations listed in Supplementary Table 1) using cDNA attached to oligo (dT)25 Dynabeads (ThermoFisher Scientific) as a template. The resulting PCR products were cloned into the pJet1.2 (ThermoFisher Scientific) or pGEMT-easy (Pro-mega), transformed into DH5α *E. coli* and the plasmid DNA was sequenced. Intron/exon boundaries were confirmed by performing a PCR using both cDNA and genomic DNA as a template, Q5 DNA polymerase (NEB) with primer pairs covering the entire predicted coding region (primer combi-nations listed in Supplementary Table1).

### Plasmid construction of GFP-fusion proteins
The final DNA sequence of all plasmids was confirmed by DNA sequencing.

**pCa_rpl44_GFP.** The *eGFP* gene was amplified by PCR using the sense primer 5'-ATTC**TAC GTA**G<u>CATGC</u>*TCTAGA*ATGGTGAGCAAGG GCGAGGAG-3' (SnaBI site bold, SphI site underlined, XbaI site italics) and the antisense primer 5'-ATTC**GCGGCCGC** TTACTTGTA-CAGCTCGTCCATG-3'(NotI site bold). The resulting PCR product was digested with SnaBI and NotI and cloned into the SnaBI and NotI sites of pCa_rpl44[27]. The resulting plasmid was termed pCa_rpl44_GFP (Sup-plementary Fig. 10).

**Actin.** The *Ca_rpl44* promoter sequence was PCR amplified from the plasmid pCa_rpl44[27] using the sense primer 5'-ATT GGG TAC CGG GCC CCC CGG TCC GTC TGT TCT GTT GTG-3' and the antisense primer 5'- TGC TCA CCA TGT ACT TTG CAA TTC AAT TCA AGA TAC-3'. The *eGFP* Gene was PCR amplified from pCa_fcp_GFP[27] using the sense primer 5'-TGC AAA GTA CAT GGT GAG CAA GGG CGA G -3' and the antisense primer 5'- CGT CAG ACA TAC CAC CAC CAC CCT TGT ACA GCT CGT CCA TGC-3'. The *C. australis actin* gene (*Ca_actin_2*) was PCR amplified from genomic DNA using the sense primer 5'- GGT GGT GGT GGT ATG TCT GAC GAC GAA GAT ATC-3' and antisense primer 5'- TTG CGC GGC CTT AGA AGC ACT TGC GGT G-3'. The *Ca_rpl44* terminator sequence was PCR amplified plas-mid pCa_rpl44[27] using the sense primer 5'-GTG CTT CTA AGG CCG CGC AAA CCA AAC G-3' and the antisense primer 5'- CAG AAC AGA CGG ACT GCA GGA GCT CCG ATG GCA ACA GC-3'. The pCa_rpl44/nat plasmid[27] was digested with XhoI and EcoRI, and was assembled with the PCR fragments in NEBuilder® HiFi DNA assembly reaction (NEB) and transformed into DH5α *E. coli*. The resulting final plasmid was termed pCa_rpl44_GFP$_{gly4}$_actin + Ca_rpl44/nat (Sup-plementary Fig. 11).

**CaMyo51A.** The *CaMyo51A* gene was PCR amplified from genomic DNA using the sense primer 5'-ATAC **TACGTA**ATGGAAGA-TATCAAGTCCACAG-3' (SnaBI site bold) and the antisense primer 5'-GTTCCCTCTCC TCCTACTTTTCA**TCTAGA**ATTC-3' (XbaI site bold). The resulting PCR product was digested with SnaBI and XbaI and cloned into the SnaBI and XbaI sites of pCa_rpl44_GFP. The resulting plasmid was termed pCa_rpl44_CaMyo51AGFP (Supplemen-tary Fig. 12).

**CaMyo51B.** The *CaMyo51B* gene and promoter sequence (1,000 bp) were PCR amplified from genomic DNA using the sense primer 5'-GGC GAA TTG GGT ACC GGG CCC CCC TCG AGC ATT CAG CTG AAT

GAA GGA TTT CAT G-3' and antisense primer 5'-CTT GCT CAC CAT AGC TAC GGA TTC GGC AGC-3'. The *eGFP* Gene was PCR amplified from pCa_fcp_GFP[27] using the sense primer 5'-CGA ATC CGT AGC TAT GGT GAG CAA GGG CGA G-3' and the antisense primer 5'-CGT CCC ATG AAC TTT ACT TGT ACA GCT CGT CCA TG-3'. The *CaMyo51B* terminator sequence (500 bp) was PCR amplified from genomic DNA using the sense primer 5'-CTG TAC AAG TAA AGT TCA TGG GAC GCC ACC-3' and the antisense primer 5'-CAC AAC AGA ACA GAC GGA CTG CAG GAG CAG GCT CAG CTG CGT G-3'. The pCa_rpl44/nat plasmid[27] was digested with XhoI and EcoRI, and was assembled with the PCR fragments in NEBuilder® HiFi DNA assembly reaction (NEB) and transformed into DH5α *E. coli*. The resulting final plasmid was termed pCaMyo51BGFP+ Ca_rpl44/nat (Supplemen-tary Fig. 13).

**CaMyo51C.** The *CaMyo51C* gene and promoter sequence (1000 bp) were PCR amplified from genomic DNA using the sense primer 5'-GGC GAA TTG GGT ACC GGG CCC CCC TCG AGG GCA CTC CTG GCA ACA TTC GTA CTA G-3' and antisense primer 5'-CTT GCT CAC CAT CAG GCC TGG GGC GGA TGC -3'. The *eGFP* Gene was PCR amplified from pCa_fcp_GFP[27] using the sense primer 5'-CGC CCC AGG CCT GAT GGT GAG CAA GGG CGA G-3' and the antisense primer 5'-CTG CTG CCG TCG CTT ACT TGT ACA GCT CGT CCA TG-3'. The *CaMyo51C* terminator sequence (500 bp) was PCR amplified from genomic DNA using the sense primer 5'-CTG TAC AAG TAA GCG ACG GCA GCA GTA TTT TC-3' and the antisense primer 5'-CAC AAC AGA ACA GAC GGA CTG CAG GCA GAT CTG CAC CAA GAA GAA C-3'. The pCa_rpl44/nat plasmid[27] was digested with XhoI and EcoRI, and was assembled with the PCR fragments in NEBuilder® HiFi DNA assembly reaction (NEB) and transformed into DH5α *E. coli*. The resulting final plasmid was termed pCaMyo51CGFP+ Ca_rpl44/nat (Supplementary Fig. 14).

**CaMyo51D.** The *CaMyo51D* gene was PCR amplified from genomic DNA using the sense primer 5'-AAA CAA CAA TTA CAA CAA CCT ACA TGC AAA GGG AAA AGGAC -3' and the antisense primer 5'-CCT CGC CCT TGC TCA CCA TTC TAG AAT CGG AAT CAG AGT CCG AG -3'. The pCa_fcp_GFP plasmid[27] was digested with SnaBI and XbaI, and was assembled with the PCR fragments in NEBuilder® HiFi DNA assembly reaction (NEB) and transformed into DH5α *E. coli*. The resulting plasmid was termed pCa_fcp_CaMyo51DGFP (Supplemen-tary Fig. 15).

### Genetic transformation of *C. australis*
Biolistic transformations to introduce genes for GFP fusion proteins were performed using a previously published protocol[27]. Briefly, $1 \times 10^8$ wild-type cells were plated on a 5 cm circle in the middle of a 1.5% EASW agar plate. Tungsten particles coated with plasmid DNA were then delivered into the cells using the Biorad PDS-1000/He™ Biolistic Particle Delivery System (28 mmHg vacuum at 1550 psi rupture). For the plasmids pCaRpl44_-Myo51A_GFP and pCafcp_Myo51D_GFP the transformations were per-formed as a co-transformation together with the plasmid pCaRpl44_nat[27] for selection using nourseothricin. Cells were scraped from the plate and grown in liquid culture medium for 24 hours and then $5 \times 10^6$ cells were plated onto 1.5% EASW agar plates supplemented with 450 μg·mL$^{-1}$ nourseothricin. Single colonies became visible following 10-14 days and were transferred into liquid EASW medium supplemented with 450 μg·mL$^{-1}$ nourseothricin.

### Spinning Disk/Laser Scanning Confocal Fluorescence Microscopy
The immobilization of cells used for confocal microscopy was achieved through one of two methods: i) 10 μL of a diatom cell suspension was pipetted on a glass-bottomed cell culture dish (Ibidi, Gräfelfing, Germany, Cat. No. 80827), and overlaid with a thin slice of 1% (w/v) agarose in EASW,

ii) 300 μL of cell suspension were pipetted into a glass-bottomed 8-well cell culture dish (Ibidi, Cat. No. 81156) and left to settle for 45 min at 18°C under constant light. The medium was then removed gently using a micropipette, and replaced with 300 μL of 1% (w/v) low melting (gelling temperature: 25 ± 5 °C) temperature agarose (Fisher Bioreagents, Cat. No. BP165) prepared in EASW.

Laser scanning and spinning disk confocal microscopy were performed at the Light Microscopy Facility and Molecular Imaging and Manipulation Facility respectively, both Core Facilities of the CMCB Technology Platform at TU Dresden. Laser scanning confocal fluorescence microscopy (LSCM) was performed using an LSM 780/FLIM inverted microscope (Zeiss, Jena, Germany) equipped with 32-channel GaAsP spectral detectors and using a Plan-Apochromat 63x/1.46 Oil Corr M27 (Zeiss) and the Zen software (2011 version, Zeiss). Images were captured using a simultaneous three channel mode to acquire fluorescence of GFP (excitation: 488 nm, emission: 489–524 nm), chloroplast autofluorescence (excitation: 633 nm, emission: 667/20 nm), and bright-field images. Spinning disk confocal microscopy (SDCM) was performed using a Nikon Eclipse Ti-E inverted microscope equipped with an Andor iXon Ultra 888, Monochrome EMCCD camera and a 100x/1.49 SR Apochromat Oil Objective (NIKON) and NIS Elements software (version 4.5). Two laser lines were used to detect the fluorescence of GFP (excitation: 488 nm, emission: 525/30 nm) and chloroplast autofluorescence (excitation: 647 nm, emission: 685/40 nm).

### Cell Population Motility Assay

Cell lines were maintained in liquid EAWS medium in T25 cell culture flasks (Greiner, 690195) at a light intensity between 40 and 60 μmol photons·$m^{-2}$·$s^{-1}$, using a cool white light source, with a 14/10-hour light/dark cycle. Cell cultures were subcultured weekly by scrapping the cells from the bottom of the flask with a cell scraper and transferring 0.5 mL to a new flask. For the motility assay, two days after subculturing the cells were imaged at 24°C using a bright field microscopy setup on a Zeiss AxioObserver Z1 microscope equipped with a 10x Plan-APOCHROMAT, NA 0.45 lens, and a TL LED lamp set to 8.4% intensity. Movies were taken using an Axiocam 506 camera (Zeiss) camera with an effective pixel size of 0.91 μm, at a rate of 1 fps with an exposure time of 10 ms for 5 min using Zen 3.0 (blue edition) software (Zeiss). The positions of the cells in the movies were determined using the tracking software FIESTA v1.6[70]. Using the cell positions, the distance cells moved over the time of observation were determined and mean velocities were calculated for each cell. For each mutant the distributions of mean velocities are presented along with a boxplot showing the median and the IQR (interquartile range). Only cell traces that were longer than 30 s and in which cells had moved more than 50 μm (approximately one cell length) were included in the velocity distributions.

### Total Internal Reflection Fluorescence (TIRF) Microscopy

TIRF Microscopy was performed on a Nikon Eclipse Ti2 microscope equipped with a perfect focus system (PFS), a 100X, 1.49 NA oil, apochromatic TIRF objective with either a 1x or 1.5x Optovar tube lens (pixel sizes: 130 × 130 nm and 86×86 nm respectively, with 1:1 aspect ratio). Samples were illuminated with a 488 nm laser placed in a visitron laser box and channeled through an iLas2 ring TIRF module operated in ellipse mode (ring TIRF). Images from different emission channels were acquired with separate EMCCD cameras (iXon Life EMCCD for 525/30 nm (GFP) and iXon Ultra EMCCD for ≥653 nm (chloroplast) channel) each containing 1024 ×1024 pixel sensor and controlled with VisiView software. Images were acquired in time-lapse mode every 100 ms with 100 ms exposure (10 frames per second). Movies were acquired for a total time of between 1-2 minutes.

### Data processing and analysis

Movies from the VisiView software were converted into .tiff format and processed using Fiji & MATLAB. Fiji commands and plugins used are shown in *italics*.

**Tracking**. Regions of movies covering paths of single cells were cropped out either by defining a rectangular region of interest (ROI) and using *crop* or *duplicate*, or by drawing a line ROI covering the entire path followed by the cell, and then using the *straighten* command. Far-red channel data (showing chloroplast autofluorescence) was used together with *TrackMate*[71,72] to obtain the X-Y positions over time, for a chloroplast signal in the leading and trailing half of the cell (Supplementary Fig. 5b). Fixed spot sizes were set for each individual cell to match bounding circles with the diameter of the larger chloroplast (~4.5–6.0 μm, cyan and magenta open circles, Supplementary Fig. 5b). Custom MATLAB code was used to determine the angle of the long axis of the cell with respect to the lab coordinate system using the vector connecting the two bounding circles. The center position of the cell (red-filled circle) over time was defined as the geometric mean between the center positions of the two bounding circles (cyan and magenta-filled circles in Supplementary Fig 5c). Instantaneous velocity values were obtained by dividing the frame-to-frame displacement of the cell's center point by the time interval between consecutive frames (100 ms). All velocity graphs were generated in MATLAB and smoothed using the "smooth" function with a moving average of 20 data points to reduce noise caused by the time resolution of frame-to-frame tracking.

**GFP-channel registration**. A custom macro was written in Fiji to register GFP channel data. As a reference, a frame was selected in which the cell was oriented horizontally and located approximately in the center of the field of view (FOV). With respect to the orientation of the cell in this frame, all remaining frames were registered by *translate* and *rotate*, using the positions and angles calculated from the chloroplast channel as described above.

**Kymograph generation**. Kymographs of each horizontal row of pixels in registered GFP-channel stacks were generated using the *reslice* command (default parameters). These kymographs were then combined by generating a maximum projection via the *Z Project…* command in the Stacks menu. Combining these kymographs allowed us to better visualize the paths of fluorescent entities across the width of the cell (Supplementary Fig. 5d). Interference patterns and sudden changes in the z-position of the cell sometimes generated horizontal stripes of varying intensity, unrelated to the movement of fluorescently labeled entities. We removed these stripes using Fourier filtering. We performed a fast Fourier transformation of the image (*FFT*), assigned intensity values of 0 to a 2 pixel wide mask covering the height of the transformation, and back-transformed it using *Inverse FFT*.

**Myosin velocity measurements**. We employed two different methods to measure the velocities of myosin spots from kymographs. For kymographs containing a large number of spots (with similar slopes), the *Directionality* plugin was used. To do so, we selected kymograph regions in which the cell showed smooth, sustained gliding (>0.5 μm·$s^{-1}$ for at least 5 s). Rectangular ROIs (of 5-15 s) in the leading and trailing halves were drawn separately, such that they cover the paths of myosin spots (Supplementary Fig 5f, indigo, red, cyan and yellow rectangles). For each ROI, the *Directionality* plugin (method: Fourier Components) was used to generate a histogram of orientations of the paths of myosin spots. We used one of three values to estimate the mean myosin velocities in the ROI: (I) the peak of the Gaussian distribution of angles determined by the directionality plugin (most common), (II) the local maximum in the histogram of angles corresponding the paths of myosin (if the goodness of fit is poor), or (III) the manually measured angles of traces identified in the orientation map (if only a small portion of the predicted angles fit the traces in the kymograph). For kymographs containing a low number of spots (with variable slopes) velocities for straight segments were estimated using the *Straight Line* tool. The orientation of each segment was used to calculate myosin spot

velocity. The above-calculated velocities were then plotted against the average cell velocity calculated for the corresponding time window (Fig. 4a).

## Reporting summary

Further information on research design is available in the Nature Portfolio Reporting Summary linked to this article.

## Data availability

The genomic DNA and protein sequences have been deposited at GenBank: CaMyo51A: PP083301; CaMyo51B PP083302; CaMyo51C: PP083303; CaMyo51D: PP083304 Source data underlying Supplementary Fig. 4 can be found in Supplementary Data 1. Source data underlying Supplementary Fig. 7 can be found in Supplementary Data 2. Source data underlying Fig. 4 can be found in Supplementary Data 3. All raw data and code used to generate the figures in this paper can be found in the following Zenodo record: https://zenodo.org/records/13343811 (https://doi.org/10.5281/zenodo.10722974).

## Code availability

Modifiable versions of Fiji and MATLAB code used to analyze microscopy data can be found in the following repository: https://github.com/metingd/DiatomMotilityMyosin (https://zenodo.org/doi/10.5281/zenodo.13737426). Code used to generate the phylogenomic tree can be found in the Zenodo record shown above.

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

## Acknowledgements
We thank Jennifer Klemm, Corina Bräuer, and Martina Lachnit for technical help. This work was supported by the Light Microscopy Facility and the Molecular Imaging and Manipulation Facility, Core Facilities of the CMCB Technology Platform at TU Dresden. We acknowledge financial support from the Deutsche Forschungsgemeinschaft (PO 2256/1-1) to N.P., (KR 1853/9-1) to N.K. R.H. was supported by the Deutsche Forschungsgemeinschaft under Germany's Excellence Strategy – EXC-2068 – 390729961- Cluster of Excellence Physics of Life at TU Dresden.

## Author contributions
N.P., N.K. and S.D. conceived the project. N.P., N.K., S.D., M.G.D., V.F.G., L.N. and R.H. conceptualized and designed the experimental research and analyses. M.G.D., N.P. generated the DNA constructs. M.G.D., L.N. and N.P. performed the microscopy experiments. N.P., M.G.D. and V.S. determined the myosin gene models, M.G.D., V.F.G., L.N., M.L.Z. and J.R.S. wrote code for analyses and analyzed the data. M.G.D., N.P., N.K. and S.D. wrote the manuscript. All authors discussed the data, reviewed and edited the manuscript.

## Funding

## Competing interests

The authors declare no competing interests.
