## [Peer review file · Communications Biology]

Reviewers' comments:

Reviewer #1 (Remarks to the Author):

Raphid diatoms exhibit gliding motility that has been proposed to be actomyosin based, using a mechanism similar to Apicomplexans such as *Plasmodium* and *Toxoplasma*. Davutoglu et al employ live cell TIRF imaging of actin and myosins in stationary and moving *Craspedostauros australis* to gain insight into the mechanism of raphid gliding motility. Transgenic strains expressing a GFP-actin fusion show localization to the two raphe-associated actin bundles (RA-actin) that appear to be continuous. Simultaneous imaging of actin and chloroplast autofluorescence (to track cell movement) showed that actin remains fixed relative to cell movement, suggesting that actin dynamics do not play a role in gliding motility. A search for myosin genes uncovered a subset that are present only in raphid diatoms, four Class 51 myosins - MyoA-D. The behavior of each myosin in motile *C. australis* was then characterized by imaging of myosin-GFP fusions. MyoA moves quite slowly, intermittently and bidirectionally - its motion is not obviously coordinated with cell movement. In contrast, MyoB-D punctae move in the opposite direction of the cell at twice the speed of cell movement. The myosin punctae can rapidly change direction when the cell does. The results provide the first glimpse into the *in vivo* movement of raphid myosins and the relationship of their motility to gliding motility of the cell.

The myosins of the larger group of Apicomplexans that encompasses diatoms have been woefully understudied, with most work being limited to cataloging sequences, carrying out phylogenetic analysis and some localization in fixed cells. Tracking of raphid diatom myosins during different phases of gliding motility uncovers interesting myosin behaviours and is consistent with a role for MyoB-D in force generation during *C. australis* gliding motility. These results represent an important step towards better characterization of this family of motors and how they might contribute to raphid gliding motility. The findings lead to a number of compelling questions to be addressed in the future. Central among these is what is the organization of the RA-actin bundle? Are the filaments bidirectional or is there a continuous unidirectional filament bundle that runs around the perimeter of the cells as is seen for actin bundles in Characean algae? Is it possible that one or more of the *C. australis* myosins exhibits 'reverse' directionality? How do the different myosins operate together (or do they)? Overall, this really nice work that sets the stage for making progress in characterizing another mode of gliding motility that will be of interest to workers in the field of cell biology and cellular motility.

Detailed comments -

1) It would be helpful if the authors could include a diagram of the major morphological features of the raphid diatom so that readers who are unfamiliar with these awesome organisms can orient themselves to the main features of the diatom where actin and myosin are localized. For example, many may not know what the valve is, leading to some confusion when localization to this structure is discussed.

2) CaMyoA-D are all class 51 myosins (pg 5) - the myosins should be referred to as CaMyo51A, Myo51B, etc., as per convention in the field.

While it is not the focus of this paper, the authors characterized the full complement of myosins from *C. australis* and some basic information about all of these should be provided (perhaps in the supplement). It is not enough to say that the family is similar to that reported for myosins in the related species *P. tricornutum* and *T. pseudonana* (pg. 5).

How many actin genes are present in *C. australis* and are they highly identical? If there are multiple actins it known/possible that there could be differential localization of actins in this organism?

3) A phylogeny of myosins from a number of diatom species is presented (Supp. Fig 4). It is not readily apparent how many different species are represented in the Cladogram. These should all be listed, either in the Methods or in a supplementary file. It would be helpful if the authors could provide links to the genome databases that were used to identify the various myosin genes. The alignments used for the phylogeny should be available to readers.

(It appears that some or most of this information has been uploaded to Zenodo but while this reviewer was able to download the Zip file it was not possible to open the individual files to determine what information was being provided.)

The authors could consider providing an alignment of the core motor domains along with the sequences of a small number of more familiar crystallized myosin motor domains to orient the reader (e.g. rabbit skeletal myosin II, *Dictyostelium* myosin II, *Dictyostelium* Myo1E, etc). This would highlight the similarities and possible differences in the myosin motor domain sequences for the myosins being studied here.

4) The authors say that they observe inhomogeneities in the GFP myosin signals either while cells are stationary or are gliding (pg 6). It would be helpful to refer readers to Supp Fig 5 where this is seen more clearly than in main Fig 3.

5) Supplementary Fig 6 nicely illustrates the dynamics of the different myosin during different phases of motility. These key data should be moved into the main text of the

paper.

6) Figure 4 is trying to convey two different, although quite related points. One is the velocity of the myosins and the other is the velocity of the cell that the myosin is moving in.

The plot in Fig 4a is difficult to understand at first glance. The seemingly random location of the data points for MyoB-D make it hard to know what is being conveyed especially since the x-axis is not clearly labeled on the bottom.

Myosin velocities are typically plotted on histograms that shows the mean velocity for a given motor. Was this done for each myosin?

The Nmeasurements seems rather low, making it difficult to be fully confident of the values reported here - is that number of velocities measured or number of cells analyzed?

7) The description of the generation of the expression plasmids could be clearer. It seems that expression is being driven by a ribosomal promoter, is that the case? What is the terminator? In the case of cloning the genes some appear to be derived from genomic DNA that includes promoter and terminator sequences and then the coding regions are subcloned. Is that case? Do those genes have introns in them? In other cases it seems that cDNA may have been used. A general overview of the approach that includes summary of the base expression plasmid and then how the genes were assembled into should be provided at the beginning.

8) Top of pg 17 - should it be Supplementary 2d?

Reviewer #2 (Remarks to the Author):

Review of Davutoglu et al., Gliding motility of the diatom *Craspedostauros australis* correlates with the intracellular movement of raphid-specific myosins.

The authors did a study on the intracellular contribution of actomyosins to gliding motility via the raphe in the diatom *Craspedostauros australis*. They found that gliding motility was induced by intracellular myosins (CaMyoB-D), not actin, as the inverse forcing molecular mechanism in this diatom. This is a novel result. The authors presented a clear, detailed, and well-written manuscript. The level of detail in the text is quite good as are the figures, videos, and other supplementary materials. What is promised is what is delivered. Unknowns are acknowledged. This manuscript will be of great interest to others in diatom research and other biological disciplines and related fields such as biofouling, nanoscience, materials engineering, biosensing, or physical

biology.

A few minor changes as well as questions and comments come to mind that are meant to be thought-provoking and could be helpful to round out the discussion in the manuscript.

Changes, questions, and comments:

1. Change the terms “correlates with,” “correlation,” or other versions of this word in the title and text. Someone may get the impression that you did least-squares or regression statistical analyses when you did not. The terms “corresponds to” and “correspondence” would be more appropriate.

2. A couple of typos: p. 2, last paragraph, line 4 – “exisits” should be “exists;” Reference 43 – remove “V.”

3. Bead/particle streaming and elastic snapping are used to describe gliding/jerky motion and reversals. Would you say that this system is random or chaotic in some parts, while other aspects of the system are deterministic? If random, is this Brownian motion or Gaussian noise? If chaotic, how would you show that this is true?

4. What type of bead (e.g., polystyrene or other polymer or non-polymeric material) is best to mimic gliding motion in diatoms? How does bead size or shape affect the force necessary to induce particle streaming that induces diatom gliding motion?

5. Is stiffness a consideration with respect to bead streaming, and therefore diatom gliding? Deformation with respect to beads and streaming? If so, how does this affect diatom gliding?

6. The diatom gliding system is dynamical. How would you characterize the system in terms of linearity or non-linearity over time? How would you characterize the system in terms of states of stability or instability, especially with regard to jerky motions or reversals and elastic snapping?

7. Is snap-through buckling something to consider with diatom gliding and possible instability states? Explain.

8. Would hysteresis play a role in diatom gliding? If so, how?

9. What is the relation between intracellular influences and type of substrate that results in diatom smooth, jerky, or reversing motions?

Reviewer #3 (Remarks to the Author):

In their manuscript, authors perform a study of the myosins of the raphe diatom species using phylogenetic analysis and the localization of these proteins within the cell in connection with the cell locomotion movement along the substrate. The authors raised an interesting question about the cellular motility of diatoms, which has attracted the attention of researchers in various fields.

In this study, the authors used a number of modern approaches, in particular live-cell imaging using GFP-labeling of the actin and some myosins, laser scanning confocal fluorescence microscopy and total internal reflection fluorescence microscopy, which made it possible to characterize some features of the diatom myosins using the example of *Craspedostauros australis*. From the variety of diatom myosins, the authors identified several myosins characteristic for raphe diatoms, CaMyoA-D. The structure of the sequences of these proteins was characterized, and their localization in living cells was revealed. In the present study, the subcellular localization of CaMyoB-D was shown to correlate with cell motility.

Below, I briefly note some points that I would like the authors to address.

1. I noticed that during phylogenetic analysis and to characterize the main domains of the studied myosins, predicted amino acid sequences were used. However, this is not mentioned in the methods and results description. The methods describe a search in available genomes and transcriptomes, and after that the authors immediately proceed to filtering the found protein sequences. It would be good to mention that authors are talking about predicted amino acid sequences in both materials and methods and results.

2. In materials and methods there is a description of transmission electron microscopy, however, the only image of a cell section is available only in supplementary materials. Moreover, the authors indicate in the results that the localization of actin coincides with the localization of Golgi (Supplementary Fig. 1). I would like to draw the authors' attention to the fact that most of the cytoplasm of the cell of the studied species (in this case the chloroplast can be ignored) is located around the nucleus and, quite naturally, actin, as part of the cytoskeleton, involved in many processes in the cell, is localized precisely in the cytoplasm. There is no particular indication in the results that the Golgi is the key factor determining actin localization. In general, if the authors did not perform immunoelectron microscopy, it is impossible to talk about the localization of any proteins based on the cell structure on the sections. In my opinion, the TEM data in the work does not provide new information and can be removed, although electron microscopy of sections for diatoms is still rare.

3. I am not entirely satisfied with the phylogenetic tree on Supplementary Fig. 4. In my opinion, the filtering of sequences was not done entirely correctly. It would be worth keeping only the most complete sequences and not using fragments in the analysis. It should be checked whether all analyzed sequences are myosins with their characteristic domains. In this case, it is likely that the myosins would be located in a more orderly manner on the tree, and it would also be possible to determine unclassified myosin sequences. In addition, 52 diatom species are mentioned, however, it is not clear where exactly which species were included in the study are described. In my opinion, such an analysis is an essential part of the study, because on its basis the selection of myosins that are described here was made. Therefore, a detailed description of how this conclusion was obtained and on the basis of what data is necessary in the paper.

4. For some unknown reason, the authors talk a lot about the mysterious forces that cause the movement of diatoms along the substrate, mention EPS threads and the composition of the mucilage secreted by diatoms, however, the well-known mechanism of interaction between actin and myosin, which provides the delivery of some vesicles to the plasmalemma, is mentioned only in the end of the discussion. The mechanism is described in many modern textbooks on cell biology, using the example, of course, of other organisms. The suggestion that actin-myosin interaction mediates the vesicular delivery and release of mucous fibers required for diatom locomotion is currently the most likely explanation for diatom gliding. The possibility of the existence of such a mechanism, based on previously known data, has already been mentioned in one of the books that the authors refer to (I will not name the exact article, since I do not want to tell the authors what works they should cite in their article). All the results obtained by the authors of this work indicate the existence of this mechanism. However, the authors seem to be embarrassed to write clearly about this.

In general, the study was carried out at a high level and needs to be published as soon as possible, since the data obtained are new and will be of interest to many researchers. However, the description of such work requires precision and accuracy, as well as a comprehensive discussion involving data of modern cell biology. I hope that the authors will be able to take into account the reviewer's recommendations, in which case the work will be presented in the most complete and comprehensive manner.

Reviewer #4 (Remarks to the Author):

This paper presents in vivo data on the dynamics of actin and myosin in gliding diatoms; it is one of the first experimental tests of the actomyosin-based adhesion motility complex (AMC) model—a well-known proposed mechanism for diatom motility. Four raphid diatom-specific myosins in Craspedostauros australis were identified in silico, and their dynamics tested in vivo, where evidence supporting the AMC model was

found. The observed correlation between the velocity of GFP-tagged myosin motors and the velocity of the gliding diatom body, as expected by the AMC model, is the key result of this work. The work is of remarkable interest for those interested in the biology of microalgae, with a focus on the mechanism of diatom gliding—a subject of much debate and speculation. Overall, the work is systematic and presented with clarity, both in concept and in exposition. The methodologies used (computational analysis, microbiology, molecular biology, fluorescence microscopy, digital image processing) are appropriate and of excellent quality. The literature cited includes necessary and relevant references to follow this work appropriately. The discussion and conclusions address experimental observations and their relationship with what is known in the field, including the most recent literature. This reviewer recognized this effort by the authors and recommends publication with minor revisions.

General comments:

1. p. 4 “Motile cells were observed in all clones irrespective of the abundance of actin-GFP, indicating that the GFP tag did not inhibit cell motility.”

- Was this observation (motility in WT vs. actin-GFP strain) quantified?

- Is motile behavior unchanged for the myosin-GFP strains?

2. p. 4 “By effectively subtracting the cell movement from the kymograph (space-time plots of the fluorescence intensities along the direction of the raphe), we were able to relate the intracellular movement of fluorescently labeled actin (or myosin, see below) with respect to cell movement (Supplementary Fig. 2, Methods).”

For this methodology, I understand it is assumed that chloroplast positions (and possibly shape) are unchanged relative to the diatom body during an experiment. Yet, it is known that chloroplasts in diatoms do migrate—particularly upon illumination. Moreover, I wonder if fading of chloroplast autofluorescence does not impact in the registration strategy. Please comment about how you can discard these effects.

3. p. 7 “The absolute values of the associated myosin velocities (ranging up to $12 \mu\text{m s}^{-1}$) always exceeded the cell velocities (ranging up to $4 \mu\text{m s}^{-1}$)”

A possible explanation provided for this observation is mechanical compliance in the motility machinery (p. 10). Could this effect be observed in cells that transition from stationary to motile, where the start of Myo-GFP motion may not coincide with the start of diatom motion, or vice versa?

Specific comments:

1. p. 4 “apicies” change to “apices”
2. p. 9 “relavant” change to “relevant”
3. p. 10 “Ca5609; >1 mDa” change to “Ca5609; >1 MDa”
4. Suggestion: Please make the time axis of the “Cell Velocity” vs “Time” graphs match in size the time axis of kymographs. This would help in comparing cell velocity with localization of Myosin-GFP (this comparison is currently difficult in Figs. 3d, 5a, for example).
5. Fig. 4b. I wonder whether it is truly necessary to quote myosin velocities with three significant digits provided that uncertainties are in the first significant digit; e.g. $-6.00 \pm 1.46 \mu\text{m s}^{-1}$.

We thank all reviewers for their careful and constructive review of our manuscript. Please find below our detailed point-to-point responses to the reviewers' comments and our actions taken. In addition, the changes are indicated using Trackchanges, as well as blue colour (no Markup view) to highlight key changes in a submitted version of the manuscript. Where line numbers are indicated these refer to the 'All Markup' view in the revised manuscript.

In particular, we (i) performed additional experiments with regard comparing the motility of the utilized *C. australis* cell lines (Supplementary Fig. 4), (ii) added new data of the evaluation of the myosin velocities (Fig. 4b), (iii) included significantly more information on the phylogenomic analysis, myosin sequences and the employed DNA constructs (Supplementary Figs. 7, 8 and 12-15, as well as Supplementary Table 1 and Supplementary Data 1), (iv) added former Supplementary Fig. 6 into the main text (Fig. 6), and (v) extended the interpretation and discussion of our results (at various places in the manuscript).

Reviewer #1 (Remarks to the Author):

	Reviewer Comment	Response
1	It would be helpful if the authors could include a diagram of the major morphological features of the raphid diatom so that readers who are unfamiliar with these awesome organisms can orient themselves to the main features of the diatom where actin and myosin are localized. For example, many may not know what the valve is, leading to some confusion when localization to this structure is discussed.	We have added a schematic to Figure 1 to show the main features of a pennate diatom cell wall and the intracellular organization. Lines: 883-884
2	(a) CaMyoA-D are all class 51 myosins (pg 5) - the myosins should be referred to as CaMyo51A, Myo51B, etc., as per convention in the field. (b) While it is not the focus of this paper, the authors characterized the full complement of myosins from C. australis and some basic information about all of these should be provided (perhaps in the supplement). It is not enough to say that the family is similar to that reported for myosins in the related species P. tricornutum and T. pseudonana (pg. 5). (c) How many actin genes are present in C. australis and are they highly identical? If there are multiple actins it known/possible that there could be differential localization of actins in this organism?	(a) We have modified the text and figures to reflect the conventional naming of the myosins. (b) We have added an additional figure to the supplementary information (Supplementary Figure 8) to show the full complement of myosin sequences in C. australis. As the gene models for these other myosins were determined by Valeria Sabatino, she is now an additional co-author on the manuscript. The primers used for the RACE PCR of these additional C. australis myosins is included in the Supplementary Table 2. (c) Interestingly, C. australis has four actin genes that are located near one another on contig 000007F. We have added additional information to show the genomic location of these genes, a schematic of the gene structure, and alignments of both the DNA and protein sequences (Supplementary Figures 1-3). There is a single amino acid exchange in just one of the four actin sequences.

		For GFP-tagging we have cloned Ca_actin_2 under control of the constitutive rpl44 promoter sequence. Therefore, if there are for example specific cell cycle localizations for each of the four different actin genes we have not been able to see this in our experiments. Other diatoms have been reported to contain one to six copies of actin genes that are typically highly conserved (Aumeier, et al 2015. Actin, actin-related proteins and profilin in diatoms: A comparative genomic analysis. Marine Genomics, 23, 133-142). Lines: 94-96
3	(a) A phylogeny of myosins from a number of diatom species is presented (Supp. Fig 4). It is not readily apparent how many different species are represented in the Cladogram. These should all be listed, either in the Methods or in a supplementary file. It would be helpful if the authors could provide links to the genome databases that were used to identify the various myosin genes. The alignments used for the phylogeny should be available to readers. (It appears that some or most of this information has been uploaded to Zenodo but while this reviewer was able to download the Zip file it was not possible to open the individual files to determine what information was being provided.) (b) The authors could consider providing an alignment of the core motor domains along with the sequences of a small number of more familiar crystallized myosin motor domains to orient the reader (e.g. rabbit skeletal myosin II, Dictyostelium myosin II, Dictyostelium Myo1E, etc). This would highlight the similarities and possible differences in the myosin motor domain sequences for the myosins being studied here.	(a) The information associated with the phylogenetic analyses was included as part of the Zenodo dataset. We realize that this caused some issues for the reviewers and we have now included this information in the Supplementary Information (Supplementary Table 1, Supplementary Data 1) and the figure legend of Supplementary Figure 4. Perhaps the reviewer did not notice that we have provided references in the materials and methods section for the database used to retrieve the myosin sequences (see below). Nevertheless, we have now include the links and references to these databases as part of the legend for Supplementary Table 1. Priyam, A. et al, Sequenceserver: A Modern Graphical User Interface for Custom BLAST Databases, Molecular Biology and Evolution, 36, 2922–2924(2019). Keeling PJ, Burki F, Wilcox HM, Allam B, Allen EE, et al. The Marine Microbial Eukaryote Transcriptome Sequencing Project (MMETSP): Illuminating the Functional Diversity of Eukaryotic Life in the Oceans through Transcriptome Sequencing. PLOS Biology 12, e1001889. (2014) Osuna-Cruz, C.M., Bilcke, G., Vancaester, E. et al. The Seminavis robusta genome provides insights into the evolutionary adaptations of benthic diatoms. Nat Commun 11, 3320 (2020). (b) We now also provide an alignment of the CaMyo51A-D together with a Rabbit skeletal myosin II, Human myosin II, C. elegans Myo5 and Dictyostelium Myo1E, highlighting the canonical myosin sequence domains/motifs (Supplementary Figure 9).

4	The authors say that they observe inhomogeneities in the GFP myosin signals either while cells are stationary or are gliding (pg 6). It would be helpful to refer readers to Supp Fig 5 where this is seen more clearly than in main Fig 3.	The text has been modified to also refer to these other figures. Line: 190
5	Supplementary Fig 6 nicely illustrates the dynamics of the different myosin during different phases of motility. These key data should be moved into the main text of the paper.	The Supplementary Figure 6 has been moved to the main text as Figure 6.
6	Figure 4 is trying to convey two different, although quite related points. One is the velocity of the myosins and the other is the velocity of the cell that the myosin is moving in. The plot in Fig 4a is difficult to understand at first glance. The seemingly random location of the data points for MyoB-D make it hard to know what is being conveyed especially since the x-axis is not clearly labeled on the bottom. Myosin velocities are typically plotted on histograms that shows the mean velocity for a given motor. Was this done for each myosin? The $N_{\text{measurements}}$ seems rather low, making it difficult to be fully confident of the values reported here - is that number of velocities measured or number of cells analyzed?	We have now included an additional panel (Fig. 4b) with box plots showing the distribution of data points for the velocities of CaMyo51A-D. For CaMyoB-D, N_{cells} corresponds to the number of cells investigated. $N_{\text{measurements}}$ represents the total number of regions across all kymographs that were used to estimate the mean velocity of each myosin. Although this number may be low, each measurement represents an average of tens to hundreds of traces of myosin spots in the respective kymograph. We clarified this in the legend of Fig. 4. Lines: 934-941
7	The description of the generation of the expression plasmids could be clearer. It seems that expression is being driven by a ribosomal promoter, is that the case? What is the terminator? In the case of cloning the genes some appear to be derived from genomic DNA that includes promoter and terminator sequences and then the coding regions are subcloned. Is that case? Do those genes have introns in them? In other cases it seems that cDNA may have been used. A general overview of the approach that includes summary of the base expression plasmid and then how the genes were assembled into should be provided at the beginning.	We have modified the description for the generation of the expression plasmids to ensure clarity and included plasmid maps for each construct in the Supplementary information (Supplementary Figures 11-15). For all constructs, as stated in the text, we have amplified the gene from genomic DNA, therefore, when present, the expressed gene contains introns. Genomic DNA was abbreviated to gDNA, we have changed this to 'genomic DNA' in the M&M section. We appreciate that it is perhaps confusing as each of the four GFP-tagged myosins are expressed under different regulatory sequences. The reason for this was that we needed to identify cell lines where the GFP signal was strong enough to image using TIRF microscopy. In two cases, the endogenous regulatory sequences of the myosins

		were strong enough, whereas for the other two we have used constitutive promoters (Rpl44 or fcp).
8	Top of pg 17 - should it be Supplementary 2d?	Yes, this was a typo. We have corrected this in the text.

Reviewer #2 (Remarks to the Author):

	Reviewer Comment	Response
1	Change the terms “correlates with,” “correlation,” or other versions of this word in the title and text. Someone may get the impression that you did least-squares or regression statistical analyses when you did not. The terms “corresponds to” and “correspondence” would be more appropriate.	We agree the "correlate with" may implicate something we have not performed. We decided to use “coincides with” for describing the relationship between the myosin and cell movement. We have changed this term throughout the Main Text and Supplementary Information.
2	A couple of typos: p. 2, last paragraph, line 4 – “exisits” should be “exists;” Reference 43 – remove “V.”	We fixed these typos.
3	Bead/particle streaming and elastic snapping are used to describe gliding/jerky motion and reversals. Would you say that this system is random or chaotic in some parts, while other aspects of the system are deterministic? If random, is this Brownian motion or Gaussian noise? If chaotic, how would you show that this is true?	These reviewer comments (3-9) do not seem to be related to our manuscript as we have not performed experiments with beads. Although we reference the work of Gutiérrez-Medina et al. (DOI 10.1088/1478-3975/ac7d30) in our introduction and discussion, we wonder if comments 3-9 might actually pertain to this previous study rather than our own work. Hence, we did not feel in the position to address these comments.
4	What type of bead (e.g., polystyrene or other polymer or non-polymeric material) is best to mimic gliding motion in diatoms? How does bead size or shape affect the force necessary to induce particle streaming that induces diatom gliding motion?	
5	Is stiffness a consideration with respect to bead streaming, and therefore diatom gliding? Deformation with respect to beads and streaming? If so, how does this affect diatom gliding?	
6	The diatom gliding system is dynamical. How would you characterize the system in terms of linearity or non-linearity over time? How would you characterize the system in terms of states of stability or instability, especially with regard to jerky motions or reversals and elastic snapping?	
7	Is snap-through buckling something to consider with diatom gliding and possible instability states? Explain.	
8	Would hysteresis play a role in diatom gliding? If so, how?	

9	What is the relation between intracellular influences and type of substrate that results in diatom smooth, jerky, or reversing motions?	
---	---	--

Reviewer #3 (Remarks to the Author):

	Reviewer Comment	Response
1	I noticed that during phylogenetic analysis and to characterize the main domains of the studied myosins, predicted amino acid sequences were used. However, this is not mentioned in the methods and results description. The methods describe a search in available genomes and transcriptomes, and after that the authors immediately proceed to filtering the found protein sequences. It would be good to mention that authors are talking about predicted amino acid sequences in both materials and methods and results.	We improved our description with regard to this issue. For details, please see our responses to Reviewer #1. We have added the term ‘predicted’ to the results and M&M sections. Lines 149 & 408
2	In materials and methods there is a description of transmission electron microscopy, however, the only image of a cell section is available only in supplementary materials. Moreover, the authors indicate in the results that the localization of actin coincides with the localization of Golgi (Supplementary Fig. 1). I would like to draw the authors’ attention to the fact that most of the cytoplasm of the cell of the studied species (in this case the chloroplast can be ignored) is located around the nucleus and, quite naturally, actin, as part of the cytoskeleton, involved in many processes in the cell, is localized precisely in the cytoplasm. There is no particular indication in the results that the Golgi is the key factor determining actin localization. In general, if the authors did not perform immunoelectron microscopy, it is impossible to talk about the localization of any proteins based on the cell structure on the sections. In my opinion, the TEM data in the work does not provide new information and can be removed, although electron microscopy of sections for diatoms is still rare.	We agree that we do not provide any new information with these TEM images. We have therefore decided to remove this figure from the supplementary information.
3	I am not entirely satisfied with the phylogenetic tree on Supplementary Fig. 4. In my opinion, the filtering of sequences was not done entirely correctly. It would be worth keeping only the most complete sequences and not using fragments in the analysis. It should be checked whether all analyzed sequences are myosins with their characteristic domains. In this case, it is likely that the myosins	We have now modified the text to make it clear that, due to MMETSP database containing many truncated transcripts, we have only included those myosin sequences that contain the canonical myosin motor domain features (P-loop, switch I, switch II, actin-binding sites) and trimmed all of the sequences to this region of the motor

	would be located in a more orderly manner on the tree, and it would also be possible to determine unclassified myosin sequences. In addition, 52 diatom species are mentioned, however, it is not clear where exactly which species were included in the study are described. In my opinion, such an analysis is an essential part of the study, because on its basis the selection of myosins that are described here was made. Therefore, a detailed description of how this conclusion was obtained and on the basis of what data is necessary in the paper.	domain. We have double-checked the alignments and subsequently reduced the number of sequences from 320 to 309. For further details, please also see our responses to Reviewer #1. lines: 405-412
4	For some unknown reason, the authors talk a lot about the mysterious forces that cause the movement of diatoms along the substrate, mention EPS threads and the composition of the mucilage secreted by diatoms, however, the well-known mechanism of interaction between actin and myosin, which provide the delivery of some vesicles to the plasmalemma, is mentioned only in the end of the discussion. The mechanism is described in many modern textbooks on cell biology, using the example, of course, of other organisms. The suggestion that actin-myosin interaction mediates the vesicular delivery and release of mucous fibers required for diatom locomotion is currently the most likely explanation for diatom gliding. The possibility of the existence of such a mechanism, based on previously known data, has already been mentioned in one of the books that the authors refer to (I will not name the exact article, since I do not want to tell the authors what works they should cite in their article). All the results obtained by the authors of this work indicate the existence of this mechanism. However, the authors seem to be embarrassed to write clearly about this.	We thank the reviewer for suggesting we write more clearly about these ‘mysterious forces’ that propel diatom gliding. Nevertheless it would have been helpful to specifically mention the book chapter. We believe the reviewer is referring to the book ‘Diatom Gliding Motility’ edited by Cohn, Manoylov and Gordon, which contains numerous chapters discussing different hypotheses regarding the origin of the forces involved in diatom gliding. Due to this we have opted not to provide a lengthy description of these different theories in the introduction as this would have resulted in a long literature review. Towards the end of the discussion section, we mention that we believe that CaMyo51A may play a role in the intracellular vesicle delivery to the raphe. However, we do not think that the delivery and secretion of material from these vesicles alone is sufficient to drive gliding and is not consistent with the observation of bead/particle movement along the raphe. We have also modified a number of sections in the discussion to address these issues and the reviewer's concerns. Lines: 314-315 Lines: 324-328 Lines:344-349 Lines:356-373

Reviewer #4 (Remarks to the Author):

	Reviewer comment	Response
1	p. 4 “Motile cells were observed in all clones irrespective of the abundance of actin-GFP, indicating that the GFP tag did not inhibit cell motility.”  - Was this observation (motility in WT vs. actin-GFP strain) quantified? - Is motile behavior unchanged for the myosin-GFP strains? 	In the Supplementary Information (Supplementary Fig. 2) we have now included a quantification of the cell population velocities of the different GFP-tagged cell lines compared to the wild type, showing that they are consistent. These experiments are very challenging to perform as the motility of diatoms in culture are hard to control. Often, we see different velocities (and general motility behaviour) of the same cell lines (including wild type) on different days, time of day, cell densities, surfaces, sub-culturing histories, and medium compositions. Therefore, for all cell lines we see a large spread of cell velocities. However, the mean cell velocities we report here are consistent with previous studies: Poulsen, N., Hennig, H., Geyer, V.F., Diez, S., Wetherbee, R., Fitz-Gibbon, S., Pellegrini, M. and Kröger, N. (2023), On the role of cell surface associated, mucin-like glycoproteins in the pennate diatom Craspedostauros australis (Bacillariophyceae). J. Phycol., 59: 54-69. https://doi.org/10.1111/jpy.13287 Lind, J. L., Heimann, K., Miller, E. A., Van Vliet, C., Hoogenraad, N. J. & Wetherbee, R. 1997. Substratum adhesion and gliding in a diatom are mediated by extracellular proteoglycans. Planta 203: 213–21. Lines: 104-105 Lines: 197-199 Lines: 541-557
2	p. 4 “By effectively subtracting the cell movement from the kymograph (space-time plots of the fluorescence intensities along the direction of the raphe), we were able to relate the intracellular movement of fluorescently labeled actin (or myosin, see below) with respect to cell movement (Supplementary Fig. 2, Methods).” For this methodology, I understand it is assumed that chloroplast positions (and possibly shape) are unchanged relative to the diatom body during an experiment. Yet, it is known that chloroplasts in diatoms do	Thanks a lot for this comment. Rapid migration of the chloroplasts typically takes place when the chloroplasts themselves are exposed to high-intensity light of specific wavelengths. TIRFM significantly limits the intensity of laser light reaching the chloroplasts. Furthermore, the wavelength of laser light used in these experiments is poorly absorbed by the chloroplast. For these reasons, we did not see any rapid movement of the chloroplasts in the X and Y planes.

	migrate—particularly upon illumination. Moreover, I wonder if fading of chloroplast autofluorescence does not impact in the registration strategy. Please comment about how you can discard these effects.	Although some fading of chloroplast autofluorescence occurs over time, this effect is negligible on the time scales at which our TIRFM experiments were carried out. Moreover, the tracking of cell relies on the fitting of a fixed-size spot onto the area of chloroplast autofluorescence ensuring that the midpoint of this fluorescent area is tracked and these imprecisions average out even if small fluctuations in its size were to take place. A good indicator of the robustness of this method is that the fluorescence surrounding the nucleus forms a nearly perfectly vertical line in registered kymographs. We have modified the text in the Materials and Methods to: “Fixed spot sizes were set for each individual cell to match bounding circles with the diameter of the larger chloroplast (~4.5-6.0 μm, cyan and magenta open circles, Supplementary Fig. 3b).” (Lines: 577-578)
3	p. 7 “The absolute values of the associated myosin velocities (ranging up to 12 $\mu\text{m s}^{-1}$) always exceeded the cell velocities (ranging up to 4 $\mu\text{m s}^{-1}$)” A possible explanation provided for this observation is mechanical compliance in the motility machinery (p. 10). Could this effect be observed in cells that transition from stationary to motile, where the start of Myo-GFP motion may not coincide with the start of diatom motion, or vice versa?	Very good point. We indeed observe a small offset between starting/stopping/reversing of the cell and myosin activity. To relate to this observation, we have now added dotted, horizontal lines to Fig. 5 and discuss that the offset may be related to mechanical compliance in the figure legend. Lines: 952-954
4	p. 4 “apicies” change to “apices”	corrected
5	p. 9 “relavant” change to ”relevant”	corrected
6	p. 10 “Ca5609; >1 mDa” change to “Ca5609; >1 MDa”	corrected
7	Suggestion: Please make the time axis of the “Cell Velocity” vs “Time” graphs match in size the time axis of kymographs. This would help in comparing cell velocity with localization of Myosin-GFP (this comparison is currently difficult in Figs. 3d, 5a, for example).	corrected
8	Fig. 4b. I wonder whether it is truly necessary to quote myosin velocities with three significant digits provided that uncertainties are in the first significant digit; e.g. $-6.00 \pm 1.46 \mu\text{m s}^{-1}$.	We have modified the table to only show the values to one decimal point.

REVIEWERS' COMMENTS:

Reviewer #1 (Remarks to the Author):

The authors have addressed all of this reviewer's comments and the Discussion does a good job of considering other potential functions for the CaMyo51B-D myosins in gliding motility.

However, a question remains whether there is sufficient evidence supporting the view that these myosins have anything at all to do with driving gliding motility. While it seems quite likely that they do, this has not really been directly tested (yet!). This open question does not detract in any way from the beautiful work presented here but perhaps it should be mentioned that it remains to be demonstrated that the myosins are responsible for the observed gliding motility.

There are a few minor issues that should be addressed.

line 338

The authors describe some findings by Harbich but the cite ref 58 (Sellers & Veigel, 2006) at the end of the sentence, it appears the they must have intended to cite ref 57 there.

line 555

Reference is made to Supp Fig 2c, an alignment, when it should be Supp Fig 5c.

Figure 1e.

The perinuclear actin signal is quite faint, making it difficult to see.

Methods (pg 14) and Supp Fig 12.

It appears that the plasmid for expression of CaMyo51A-GFP. The description of how the pCaRpl44_CaMyo51A_GFP plasmid was made does not indicate inclusion of the selection cassette and it is missing from the plasmid map.

Reviewer #3 (Remarks to the Author):

Dear authors and editors,

I apologize for the delay in replying.

In my opinion, the authors did a good job, with the additions made, the work looks even better. Thank you for the extended discussion of the mechanisms of diatom movement,

it allows us to clarify the questions that researchers are currently facing. I will be glad to see the study in press soon.

Reviewer #4 (Remarks to the Author):

I thank the authors for their efforts in addressing all the reviewer´s comments. My previous concerns have been met and I would recommend this article for publication.

We thank all reviewers for their positive review of our revised manuscript. Please find below our detailed point-to-point responses to the reviewers' comments and our actions taken. In addition, the changes are indicated using Trackchanges in a submitted version of the manuscript.

Reviewer #1 (Remarks to the Author):

	Reviewer Comment	Response
1	However, a question remains whether there is sufficient evidence supporting the view that these myosins have anything at all to do with driving gliding motility. While it seems quite likely that they do, this has not really been directly tested (yet!). This open question does not detract in any way from the beautiful work presented here but perhaps it should be mentioned that it remains to be demonstrated that the myosins are responsible for the observed gliding motility.	We thank the reviewer for this comment and agree that it is important to indicate the still open question. We have revised the last two paragraphs of the discussion and hope that this now clearly addresses the reviewers concerns.
2	line 338 The authors describe some findings by Harbich but the cite ref 58 (Sellers & Veigel, 2006) at the end of the sentence, it appears the they must have intended to cite ref 57 there.	corrected
3	line 555 Reference is made to Supp Fig 2c, an alignment, when it should be Supp Fig 5c.	corrected
4	Figure 1e. The perinuclear actin signal is quite faint, making it difficult to see.	we have increased the brightness of this image so that the perinuclear actin signal is easier to see.
5	Methods (pg 14) and Supp Fig 12. It appears that the plasmid for expression of CaMyo51A-GFP. The description of how the pCaRpl44_CaMyo51A_GFP plasmid was made does not indicate inclusion of the selection cassette and it is missing from the plasmid map.	This was well spotted by the reviewer. Indeed this plasmid does not contain the antibiotic selection cassette. The biolistic transformation was performed as a co-transformation with the plasmid pCa_rpl44_nat. We have modified the M&M section to include this information. This was also the case for the pCafcp_Myo51D_GFP plasmid. “For the plasmids pCaRpl44_Myo51A_GFP and pCafcp_Myo51D_GFP the transformation were performed as a co-transformation together with the plasmid pCaRpl44_nat ²⁷ .”